# SF-PE: A Synergistic Fusion of Absolute and Relative Positional Encoding for Spiking Transformers

## Abstract

Positional signals in spiking neural networks (SNNs) suffer distortion due to spike binarization and the nonlinear dynamics of Leaky Integrate-and-Fire (LIF) neurons, which compromises self-attention mechanisms. We introduce **Spiking-RoPE**, a spiking-friendly relative rotary positional encoding that applies two-dimensional spatiotemporal position-dependent rotations to queries/keys prior to binarization, ensuring that relative phase kernels are preserved in statistical expectation under LIF dynamics while maintaining content integrity. Building on this core, we propose **Spiking Fused-PE (SF-PE)**, a scheme that fuses absolute CPG-based spikes with Spiking-RoPE. The resulting attention score decomposes into complementary row/column (absolute) and diagonal (relative) structures, thereby expanding the representable function space. We validate our method across two diverse domains (time-series forecasting and text classification) on Spikformer, Spike-driven Transformer, and QKFormer backbones. SF-PE consistently improves accuracy and enhances length extrapolation capabilities. Ablations on rotation bases and 1D vs. 2D variants support the design. These results establish rotary encoding as an effective, spiking-friendly relative PE for SNNs and demonstrate that fusing absolute and relative signals yields synergistic benefits under spiking constraints. Code: `https://anonymous.4open.science/r/SNN-RoPE-F6DE`.

## 1 Introduction

Spiking neural networks (SNNs) transmit information via discrete spikes that emulate biological firing, enabling event-driven computation with low energy and neuromorphic compatibility (Maass, 1997; Davies et al., 2018; Roy et al., 2019). Recent work has transplanted key Transformer components to SNNs, including spike-friendly self-attention (Zhou et al., 2022; Yao et al., 2023; Song et al., 2024; Zhou et al., 2024a). A persistent bottleneck, however, is positional encoding (PE). While self-attention inherently lacks order awareness and therefore requires PE (Vaswani et al., 2017), PE signals in SNNs suffer distortion through spike binarization and the nonlinear dynamics of Leaky Integrate-and-Fire (LIF) neurons.

Conventional continuous PEs (e.g., sinusoidal) differentiate positions through subtle embedding changes. However, thresholding operations distort this information by either nullifying these subtle differences (when inputs remain subthreshold) or drastically amplifying them (upon crossing the threshold). This fundamental incompatibility motivates the development of spiking-friendly PEs that can survive both binarization and LIF dynamics.

Through a systematic analysis of current approaches, we identify three critical gaps in the existing SNN positional encoding landscape. **Gap 1 (Theory):** Existing SNN Transformers predominantly rely on implicit, weight-based position learning and lack rigorous analysis of how positional information is preserved through binarization. **Gap 2 (Single-paradigm limits):** Absolute PE, such as CPG-PE (Lv et al., 2024), exhibits sensitivity to shifts and suffers from aliasing on long sequences, whereas relative PE (e.g., Gray/Log-PE (Lv et al., 2025)) encounters capacity constraints and distance-resolution limitations. **Gap 3 (Spatiotemporal modeling):** SNN data are inherently

spatiotemporal in nature, yet most PEs treat position as one-dimensional, thereby neglecting the separable time and sequence axes.

To systematically address these identified gaps, we propose **Spiking-RoPE**, a comprehensive solution that begins with redesigning rotary positional encoding specifically for SNNs. Spiking-RoPE applies position-dependent rotations to queries/keys prior to spike binarization, yielding relative phase kernels that are preserved in statistical expectation under LIF dynamics while maintaining content integrity. We further decouple rotations along sequence (length) and time axes to obtain 2D Spiking-RoPE, which explicitly models spatiotemporal relations. Finally, we integrate absolute and relative signals in **Spiking Fused-PE (SF-PE)** by combining absolute CPG-based spikes at the input with Spiking-RoPE within blocks. This fused scheme activates complementary row/column (absolute) and diagonal (relative) structures in attention maps (See Fig. 2 in the Appendix), thereby expanding representable function space.

To demonstrate the effectiveness and generalizability of our approach, we conduct extensive validation across two diverse domains (i.e., time series forecasting and text classification) and three established spiking backbones (i.e., Spikformer, Spike-driven Transformer, QKFormer), supplemented by comprehensive ablations on rotation bases and 1D vs. 2D variants. Our experimental results show that SF-PE consistently improves accuracy and strengthens length extrapolation capabilities across all evaluated scenarios.

**Contributions.**

- **C1, Theoretical foundation (Gap 1):** We rigorously prove that pre-spike rotary phases preserve relative phase kernels in statistical expectation under LIF (See Appendix A), thereby explaining why rotation-based PEs are inherently compatible with spike dynamics.

- **C2, Fused absolute-relative PE (Gap 2):** SF-PE systematically integrates CPG-PE (absolute) with Spiking-RoPE (relative), jointly inducing complementary row/column and diagonal attention structures.

- **C3, Native spatiotemporal PE (Gap 3):** Spiking-RoPE independently rotates along sequence and time axes to capture spatiotemporal relations that are inaccessible to 1D designs.

- **C4, Cross-domain evidence:** Consistent gains across backbones and tasks, plus ablations, establish robustness and generality across different domains (i.e., time series forecasting, text classification).

## 2 RELATED WORK

### 2.1 SNN TRANSFORMER ARCHITECTURES

The adaptation of Transformers to the SNN domain has gained significant momentum in recent years. Notable contributions include Spikformer (Zhou et al., 2022; 2024b), which pioneered the integration of LIF neurons into vanilla Transformers to create spiking self-attention mechanisms. Building on this foundation, Spike-driven Transformer (Yao et al., 2023; 2024) advanced the field by proposing more computationally efficient spike-based MatMul operations. Spikingformer (Zhou et al., 2023) proposed a spike-based residual learning framework. QKFormer (Zhou et al., 2024a) improved the binarization process of queries and keys to reduce information loss. However, all of these adopt approaches where weights indirectly learn positions without explicit PE, thereby exemplifying **Gap 1**.

### 2.2 ABSOLUTE PE FOR SNN

Among absolute PE methods designed specifically for SNNs, CPG-PE (Lv et al., 2024) represents the current state-of-the-art approach. This method leverages central pattern generator properties to assign distinct binary spike patterns to each position through frequency channel thresholding. While demonstrating spike consistency and neuromorphic compatibility, CPG-PE exhibits fundamental limitations that our work addresses: (1) **Translation sensitivity**, absolute coordinate terms in attention render it vulnerable to sequence shifts, and (2) **Long sequence aliasing**, finite period synthesis

causes pattern collisions that become increasingly severe in extended sequences, directly contributing to **Gap 2** identified in our analysis.

### 2.3 Relative PE for SNN

In contrast to the absolute PE method, relative PE approaches for SNNs focus on encoding positional relationships rather than absolute positions. Gray-PE and Log-PE (Lv et al., 2025) represent the most advanced relative PE approaches currently available for SNNs. Gray-PE approximates relative distances using Hamming distance-based discrete codes, while Log-PE employs log-scale distance buckets. However, both methods encounter critical limitations that reinforce **Gap 2**: Gray-PE suffers from (1) representation upper bounds due to bit capacity constraints $2^b$, and (2) distance ordering violations where Hamming distance fails to preserve actual distance relationships. Log-PE encounters (1) coarse distance resolution due to distant interaction binning, and (2) spatiotemporal instability during 2D extension, thereby highlighting the necessity for our **Gap 3** solution.

## 3 Preliminary

### 3.1 Notation

$T$ denotes the number of time steps, $L$ is the sequence length (number of tokens/patches), and $D$ is the feature dimension. Bold uppercase letters denote tensors, and operations apply to the last dimension unless otherwise specified. $\mathrm{BN}(\cdot)$ denotes batch normalization, and $\mathrm{SN}(\cdot)$ denotes the spike operation induced by LIF in Eq. 1.

### 3.2 Leaky Integrate-and-Fire (LIF) Neuron

In this study, we use LIF neurons (Maass, 1997) for spike binarization in SNNs. At discrete time $t$, the membrane potential update $H(t)$, spike $S(t)$, and post-reset potential $U(t)$ for input current $I(t)$ are as follows:

$$
\begin{aligned}
H(t) &= U(t-1) + \tfrac{1}{\tau}\big(I(t) - \big(U(t-1) - U_{\text{reset}}\big)\big), \\
S(t) &= \Theta\big(H(t) - U_{\text{thr}}\big), \\
U(t) &= \big(1 - S(t)\big)\,H(t) + S(t)\,U_{\text{reset}},
\end{aligned}
\tag{1}
$$

where $\tau$ is the leak time constant, $U_{\text{thr}}$ is the threshold, $U_{\text{reset}}$ is the reset potential, and $\Theta(\cdot)$ is the Heaviside step function. In this paper, $\mathrm{SN}(z)$ refers to spike output under LIF dynamics (e.g., $\Theta(z - U_{\text{thr}})$).

### 3.3 Spiking Self-Attention

Spiking Self-Attention (SSA) is a transformation of self-attention adapted for spike representations following Spikformer (Zhou et al., 2022). For spike tensor $X \in \{0,1\}^{T \times L \times D}$:

$$
\mathbf{Q}_c = \mathrm{BN}(X)\,W_Q, \quad \mathbf{K}_c = \mathrm{BN}(X)\,W_K, \quad \mathbf{V}_c = \mathrm{BN}(X)\,W_V,
\tag{2}
$$

where $W_{\{\cdot\}}$ are learnable linear mappings. The corresponding spike embeddings are:

$$
\mathbf{Q}_s = \mathrm{SN}(\mathbf{Q}_c), \quad \mathbf{K}_s = \mathrm{SN}(\mathbf{K}_c), \quad \mathbf{V}_s = \mathrm{SN}(\mathbf{V}_c).
\tag{3}
$$

Time indices are omitted for notational simplicity, and attention is computed at each time step as follows, where AttnMap is an integer matrix reflecting spike co-occurrence over feature dimensions:

$$
\mathrm{AttnMap} = \mathbf{Q}_s\,\mathbf{K}_s^{\top} \in \mathbb{N}_0^{L \times L}, \qquad \mathbf{SSA} = \mathrm{SN}\big(\mathrm{AttnMap} \cdot \mathbf{V}_s\big).
\tag{4}
$$

### 3.4 CPG-PE

CPG-PE (Lv et al., 2024) is an absolute PE that borrows the periodic firing principles of central pattern generators. For $K$ channels with different periods, at position $i \in \{0, \ldots, L-1\}$:

$$
u_k(i) = \cos\big(\omega_k i + \phi_k\big), \qquad k = 1, \ldots, K,
\tag{5}
$$

$$
g_k(i) = \Theta\big(u_k(i) - \tau_k\big) \in \{0, 1\},
\tag{6}
$$

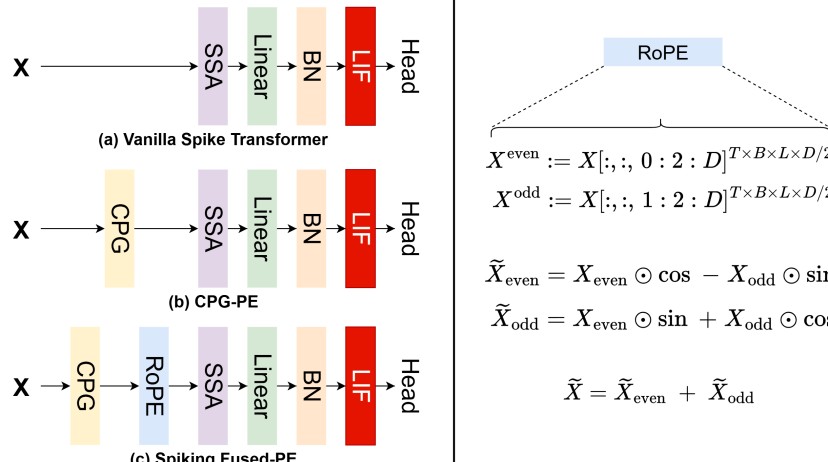

Figure 1: SFPE architecture. The diagram illustrates the integration of CPG-PE and RoPE with Spiking Neural Network, showing the flow from input spike trains through the fused PE to the attention computation in spiking transformers.

binary signals are synthesized to create:

$$\mathbf{p}_i^{\text{CPG}} = \left[ g_1(i), g_2(i), \ldots, g_K(i) \right]^{\top} \in \{0,1\}^K, \tag{7}$$

where $\omega_k$ is the angular frequency ($T_k = 2\pi/\omega_k$), $\phi_k$ is the phase, $\tau_k$ is the threshold, and $\Theta$ is the Heaviside step function.

### 3.5 ROTARY POSITIONAL EMBEDDING (RoPE)

RoPE (Su et al., 2024) encodes positional information by rotating $(2i-1, 2i)$ channel pairs of embeddings at position $m$ with position-dependent angles. For per-head dimension $d$ (even), the frequencies are set as $\theta_i = B^{-2(i-1)/d}$ ($i = 1, \ldots, d/2$), where base $B > 1$ determines the rotation frequency. The block diagonal rotation matrix at position $m \in \{0, \ldots, L-1\}$ is:

$$R_m = \text{diag}\left( \begin{bmatrix} \cos(m\theta_1) & -\sin(m\theta_1) \\ \sin(m\theta_1) & \cos(m\theta_1) \end{bmatrix}, \ldots, \begin{bmatrix} \cos(m\theta_{d/2}) & -\sin(m\theta_{d/2}) \\ \sin(m\theta_{d/2}) & \cos(m\theta_{d/2}) \end{bmatrix} \right) \in \mathbb{R}^{d \times d}. \tag{8}$$

For query/key $\mathbf{Q}_c, \mathbf{K}_c \in \mathbb{R}^{L \times d}$, RoPE is applied position-wise as:

$$(\tilde{\mathbf{Q}}_c)_m = R_m (\mathbf{Q}_c)_m, \qquad (\tilde{\mathbf{K}}_c)_m = R_m (\mathbf{K}_c)_m, \tag{9}$$

while values remain unchanged.

## 4 METHODOLOGY

### 4.1 OVERVIEW

Positional encoding (PE) in SNN-based transformers has evolved through the development process shown in Fig. 1. Early SNN transformers (a) suffered from performance degradation by learning positional information implicitly. CPG-PE (b) was introduced as the first explicit, absolute positional encoding (PE), but it revealed new limitations, such as pattern collisions in long sequences.

Building on this, our proposed **Spiking Fused-PE** (c) is a fusion approach that combines CPG-PE for learning absolute positions with our Spiking-RoPE for injecting relative relationships. This dual-stage design leverages both types of information to expand the representation space and enhance its performance.

## 4.2 SPIKING-ROPE

RoPE, originally designed for continuous neural networks, decomposes attention inner products into kernels that depend only on relative phase differences $\Delta$ through position-dependent rotations. Building on this foundation, we propose Spiking-RoPE by adapting this rotation mechanism to the pre-spike binarization stage, where relative phase kernels are maintained from a statistical expectation perspective even under LIF leakage and threshold conditions. (See Fig. 5 in the Appendix for Spiking-RoPE transformation steps; See Appendix D for the detailed implementation).

**Theoretical Foundation (Gap 1 Resolution):** The critical insight enabling Spiking-RoPE is our theoretical proof that phase rotation preserves relative positional information statistically even after spike binarization. This addresses Gap 1 by providing the first rigorous analysis of positional information preservation in SNNs. The complete theoretical analysis, including expectation preservation proofs and LIF dynamics interaction, is presented in Appendix A.

### 4.2.1 1D SPIKING-ROPE

1D Spiking-RoPE encodes relative positional information along a single axis. For this purpose, rotation matrices $R_{\varphi(i)}$ are applied to $(2r-1, 2r)$ channel pairs in even dimension $d$.

$$\tilde{q}_i = R_{\varphi(i)}\, q_i^{(c)}, \quad \tilde{k}_j = R_{\varphi(j)}\, k_j^{(c)}, \quad \Delta_{ij} = \varphi(i) - \varphi(j). \tag{10}$$

As a result, the relative relationship between two positions $i, j$ depends only on the phase difference $\Delta_{ij}$.

### 4.2.2 2D SPIKING-ROPE

To explicitly model the spatiotemporal characteristics of SNN data, 2D Spiking-RoPE encodes positional information by separating it into sequence length axis $l$ and time axis $t$. First, the embedding is divided into two equal-dimensional blocks, and independent 1D Spiking-RoPE is applied to each block. One block rotates based on sequence position $(i)$, while the other rotates based on time step $(t_i)$.

$$\tilde{q}_i = \big[\, R_{\varphi_l(i)}\, q_i^{(c,l)} \;;\; R_{\varphi_t(t_i)}\, q_i^{(c,t)} \,\big], \qquad \tilde{k}_j = \big[\, R_{\varphi_l(j)}\, k_j^{(c,l)} \;;\; R_{\varphi_t(t_j)}\, k_j^{(c,t)} \,\big]. \tag{11}$$

When computing query/key inner products, letting $\Delta_l = \varphi_l(i) - \varphi_l(j)$ and $\Delta_t = \varphi_t(t_i) - \varphi_t(t_j)$, the inner product operation naturally separates into the sum of relative phase kernel $\Delta_l$ along the length axis and relative phase kernel $\Delta_t$ along the time axis as shown below. This allows the model to consider both spatiotemporal relative distances.

$$\begin{aligned}
\langle \tilde{q}_i, \tilde{k}_j \rangle = &\big( \langle q_i^{(c,l)}, k_j^{(c,l)} \rangle \cos \Delta_l + \langle q_i^{(c,l)}, J\, k_j^{(c,l)} \rangle \sin \Delta_l \big) \\
&+ \big( \langle q_i^{(c,t)}, k_j^{(c,t)} \rangle \cos \Delta_t + \langle q_i^{(c,t)}, J\, k_j^{(c,t)} \rangle \sin \Delta_t \big).
\end{aligned} \tag{12}$$

### 4.2.3 PHASE PRESERVATION UNDER LIF

LIF performs nonlinear transformations (Eq. 1), and there is a risk of losing positional information encoded as continuous values during this process. We show that PE applied at the pre-spike stage through Spiking-RoPE is preserved from a statistical expectation perspective even after LIF's spiking transformation.

This proof involves approximating the firing probability function of LIF neurons as a linear function. Batch Normalization in SSA (Eq. 2) stabilizes the distribution of pre-spike inputs to mean 0 and variance 1. Assuming that input currents are distributed in a narrow region around the mean (0) such that the firing probability function operates almost linearly, we can show that the probability of query and key firing simultaneously in a specific dimension is approximately proportional to the product of pre-spike values $\tilde{q}_{id}, \tilde{k}_{jd}$. Under this linear approximation, the expectation of inner products over all dimensions is derived as follows.

$$\mathbb{E}[q_i^\top k_j] \approx \alpha \langle \tilde{q}_i, \tilde{k}_j \rangle, \tag{13}$$

where $\alpha > 0$ is a scaling constant depending on neuron sensitivity and input distribution, and attention scores are determined by the inner product value $\langle \tilde{q}_i, \tilde{k}_j \rangle$. When Spiking-RoPE is applied, $\langle \tilde{q}_i, \tilde{k}_j \rangle$ is expanded as a function of relative phase $\Delta_{ij} = i - j$, resulting in the final attention score having the following relationship:

$$\mathbf{SA}(q_i, k_j) \propto \mathbb{E}[q_i^\top k_j] \propto \langle q_i^{(c)}, k_j^{(c)} \rangle \cos \Delta_{ij} + \langle q_i^{(c)}, J k_j^{(c)} \rangle \sin \Delta_{ij}. \tag{14}$$

In conclusion, Spiking-RoPE preserves phase kernels containing relative positional information under nonlinear LIF dynamics, enabling the utilization of positional information in SNNs.

## 4.3 FUSED PE

In Transformer-based models, positional encoding (PE) injects order and dependencies between tokens. Existing research has independently used either absolute or relative PE. We propose fused PE, which combines both approaches to enhance positional representation power. To simultaneously reflect absolute PE $p^{\text{abs}}$ and relative PE $R_{\varphi(\cdot)}$ in input $x$, query/key are defined as follows:

$$q_i = R_{\varphi(i)} W_Q(x_i + p_i^{\text{abs}}), \qquad k_j = R_{\varphi(j)} W_K(x_j + p_j^{\text{abs}}), \tag{15}$$

where $p_i^{\text{abs}}$ is the absolute PE, and $R_{\varphi(i)} \in \mathbb{R}^{d \times d}$ is the RoPE-style block rotation at position $i$. This configuration organically fuses two information sources within a single vector by projecting content+absolute information through linear mapping, then injecting relative information through phase rotation. The row/column structure created by absolute PE and the diagonal structure created by relative PE are simultaneously activated, providing richer representation power compared to single PE (See Fig. 2 in the Appendix). Subsequently, in continuous (pre-spike) space, letting $q_i^{(c)} = W_Q(x_i + p_i^{\text{abs}})$ and $k_j^{(c)} = W_K(x_j + p_j^{\text{abs}})$,

$$\tilde{q}_i = R_{\varphi(i)} q_i^{(c)}, \quad \tilde{k}_j = R_{\varphi(j)} k_j^{(c)}, \quad \Delta_{ij} = \varphi(i) - \varphi(j).$$

With the $90°$ block rotation operator $J$ for even channel pairs, the inner product is as follows.

$$\langle \tilde{q}_i, \tilde{k}_j \rangle = \langle q_i^{(c)}, k_j^{(c)} \rangle \cos \Delta_{ij} + \langle q_i^{(c)}, J k_j^{(c)} \rangle \sin \Delta_{ij}. \tag{16}$$

According to Appendix A, this inner product approximately preserves the relative phase kernel form of the above equation in the expectation $\mathbb{E}[q_i^\top k_j]$ even after spike binarization.

## 4.4 FINAL INCORPORATION

Following fused PE, we propose Spiking Fused-PE (SF-PE), a fused method that combines CPG-PE for absolute PE and Spiking-RoPE for relative PE. First, after injecting CPG-PE from Eq. 7 into the embedding,

$$q_i^{(c)} = W_Q(x_i + E \mathbf{p}_i^{\text{CPG}}), \qquad k_j^{(c)} = W_K(x_j + E \mathbf{p}_j^{\text{CPG}}), \qquad E \in \mathbb{R}^{d \times K}, \tag{17}$$

which is rotated with 2D Spiking-RoPE. Then, the continuous inner product is decomposed with respect to the relative phases of the two axes as follows:

$$\langle \tilde{q}_i, \tilde{k}_j \rangle = \sum_{r=1}^{d_l/2} \left( A_{ij,r}^{(l)} \cos \Delta_r^{(l)} + B_{ij,r}^{(l)} \sin \Delta_r^{(l)} \right) + \sum_{r=1}^{d_t/2} \left( A_{ij,r}^{(t)} \cos \Delta_r^{(t)} + B_{ij,r}^{(t)} \sin \Delta_r^{(t)} \right), \tag{18}$$

where $d_l$ and $d_t$ denote per-axis even dimensions, and the amplitude terms are

$$A_{ij}^{(l)} = \langle q_i^{(c,l)}, k_j^{(c,l)} \rangle, \quad B_{ij}^{(l)} = \langle q_i^{(c,l)}, J k_j^{(c,l)} \rangle,$$

and similarly $A_{ij}^{(t)}, B_{ij}^{(t)}$ are defined for the time axis. Consequently, the attention score of SF-PE is structured as a sum of contributions from the length $l$ and time $t$ axes, each of which is itself a sum over individual rotational frequency channel pairs. This decomposition shows how two types of positional information are complementarily combined for each channel pair: absolute information (amplitudes $A, B$) and relative information (trigonometric kernels, $\Delta_l, \Delta_t$), allowing the model to capture richer and more granular positional details. This structure is preserved from a statistical expectation perspective even after the spike binarization process, enabling SNNs to effectively learn complex spatiotemporal patterns.

Table 1: Performance comparison on time series forecasting on 4 benchmarks with various prediction lengths 6, 24, 48, 96. The best results are shown in **bold**. PE types: A = absolute, F = Fused (absolute + relative). Metrics: higher $R^2$ and lower RSE indicate better performance. All results are averaged across 3 random seeds.

| Models | PE Type | Metric | Metr-la (L = 12) | | | | Pems-bay (L = 12) | | | | Solar (L = 168) | | | | Electricity (L = 168) | | | | Avg. |
|---|---|---|---|---|---|---|---|---|---|---|---|---|---|---|---|---|---|---|---|
| | | | 6 | 24 | 48 | 96 | 6 | 24 | 48 | 96 | 6 | 24 | 48 | 96 | 6 | 24 | 48 | 96 | |
| Transformer w/Sin-PE (Upper bound) | A | $R^2$↑ | .727 | .554 | .413 | .284 | .785 | .734 | .688 | .673 | .953 | .858 | .759 | .718 | .978 | .975 | .972 | .964 | .733 |
| | | RSE↓ | .551 | .704 | .808 | .895 | .502 | .558 | .610 | .618 | .223 | .377 | .504 | .545 | .260 | .277 | .347 | .425 | .512 |
| Spikformer w/Conv-PE Zhou et al. (2022) | A | $R^2$↑ | .713 | .527 | .399 | .267 | .773 | .697 | .686 | .667 | .929 | .828 | .744 | .674 | .959 | .955 | .955 | .954 | .733 |
| | | RSE↓ | .565 | .725 | .818 | .903 | .514 | .594 | .606 | .621 | .272 | .426 | .519 | .586 | .373 | .371 | .379 | .382 | .541 |
| Spikformer w/CPG-PE Lv et al. (2024) | A | $R^2$↑ | .726 | .526 | .418 | .287 | .780 | .712 | .690 | .666 | .937 | .833 | .757 | .707 | .972 | .970 | .966 | .960 | .744 |
| | | RSE↓ | .553 | .720 | .806 | .890 | .508 | .580 | .602 | .622 | .257 | .420 | .506 | .555 | .299 | .310 | .314 | .355 | .519 |
| Spikformer w/SF-PE (Ours) | F | $R^2$↑ | **.739** | **.561** | **.432** | **.317** | **.783** | **.713** | **.698** | **.670** | **.939** | **.877** | **.782** | **.752** | **.981** | **.975** | **.972** | **.965** | **.760** |
| | | RSE↓ | **.538** | **.698** | **.795** | **.871** | **.499** | **.576** | **.593** | **.618** | **.251** | **.362** | **.479** | **.511** | **.240** | **.280** | **.300** | **.336** | **.497** |
| SDT-V1 w/Conv-PE Yao et al. (2023) | A | $R^2$↑ | .588 | .364 | .236 | .121 | .674 | .668 | .658 | .639 | .922 | .837 | .732 | .685 | .958 | .951 | .946 | .939 | .682 |
| | | RSE↓ | .692 | .841 | .935 | .984 | .599 | .605 | .616 | .637 | .281 | .405 | .533 | .584 | .367 | .389 | .412 | .430 | .582 |
| SDT-V1 w/CPG-PE Lv et al. (2024) | A | $R^2$↑ | .601 | .387 | .257 | .152 | .693 | .695 | .680 | .664 | .935 | .860 | .748 | .710 | .966 | .955 | .959 | .945 | .700 |
| | | RSE↓ | .667 | .827 | .910 | .972 | .580 | .578 | .592 | .607 | .260 | .383 | .515 | .553 | .329 | .378 | .362 | .417 | .558 |
| SDT-V1 w/SF-PE (Ours) | F | $R^2$↑ | **.703** | **.470** | **.296** | **.187** | **.741** | **.700** | **.686** | **.679** | **.945** | **.871** | **.794** | **.766** | **.979** | **.971** | **.971** | **.969** | **.733** |
| | | RSE↓ | **.576** | **.769** | **.886** | **.952** | **.533** | **.573** | **.587** | **.593** | **.242** | **.369** | **.466** | **.496** | **.259** | **.299** | **.302** | **.310** | **.513** |
| QKFormer w/Conv-PE Zhou et al. (2024a) | A | $R^2$↑ | .706 | .509 | .411 | .275 | .735 | .671 | .667 | .663 | .927 | .841 | .737 | .689 | .966 | .961 | .958 | .955 | .729 |
| | | RSE↓ | .577 | .743 | .816 | .901 | .557 | .621 | .625 | .629 | .275 | .402 | .527 | .569 | .302 | .324 | .340 | .358 | .535 |
| QKFormer w/CPG-PE Lv et al. (2024) | A | $R^2$↑ | .711 | .522 | **.423** | .286 | .743 | .684 | .681 | **.668** | .930 | .856 | .755 | .732 | .977 | .968 | .966 | **.959** | .734 |
| | | RSE↓ | .567 | .729 | **.801** | .890 | .548 | .608 | .611 | **.623** | .271 | .389 | .508 | .531 | .264 | .289 | .307 | **.361** | .533 |
| QKFormer w/SF-PE (Ours) | F | $R^2$↑ | **.717** | **.520** | .419 | **.292** | **.749** | **.702** | **.698** | **.668** | **.934** | **.868** | **.793** | **.737** | **.981** | **.972** | **.968** | .954 | **.748** |
| | | RSE↓ | **.561** | **.730** | .804 | **.887** | **.542** | **.590** | **.594** | **.623** | **.264** | **.372** | **.468** | **.526** | **.244** | **.299** | **.318** | .383 | **.513** |

## 5 EXPERIMENTS

We evaluate on two diverse domains, time series forecasting and text classification, to test the modality-agnostic nature of SF-PE. The choice follows directly from the method's characteristics: (1) pre-spike rotary phases preserve relative kernels under LIF (C1; Sec. 4.2, Sec. 4.2.3); (2) the fused absolute–relative scheme induces complementary row/column vs. diagonal attention structure that any ordered data exhibits (C2; Eq. 16); and (3) the 2D variant decouples length and time to model spatiotemporal relations while remaining compatible with 1D sequences (C3; Sec. 4.2.2, Eq. 12). We therefore assess robustness across (a) modalities (continuous signals vs. discrete tokens) and (b) SNN backbones (Spikformer, SDT-V1, QKFormer), and we include length extrapolation to specifically probe relative-position generalization.

Our experimental validation systematically demonstrates how our gap-targeted solutions (C1-C3) translate to performance improvements across diverse domains. For our primary comparisons, we evaluate against two baselines: Conv-PE (Zhou et al., 2022; Yao et al., 2023; Zhou et al., 2024a), where positional information is learned implicitly, and CPG-PE (Lv et al., 2024), the state-of-the-art absolute PE for SNNs. However, there has been no research effort for applying relative PE to SNNs, making a direct comparison with a pre-existing method challenging (See the alternative comparison in Appendix E). Additionally, we utilize our Spiking-RoPE as a relative-only baseline and provide a detailed comparison in the ablation studies. Detailed experimental settings, including datasets, metrics, and hyperparameters, are provided in Appendix C.

### 5.1 TIME SERIES FORECASTING

Table 1 shows the performance of SF-PE on four time series forecasting datasets. The results reveal several notable patterns:

**Consistent superiority of SF-PE:** SF-PE consistently outperforms absolute PE approaches across all backbone models (Spikformer, SDT-V1, and QKFormer). In particular, the average $R^2$ score improved from 0.744 to 0.760 on Spikformer and showed a substantial improvement from 0.700 to 0.733 on SDT-V1. Similarly, SF-PE achieved a leading average $R^2$ score of 0.748 on QKFormer.

**Robustness in long-term prediction:** While the performance degradation occurs as prediction length increases (6 to 96 hours), SF-PE maintains relatively stable performance compared to other methods. Specifically, in the 96-hour prediction on the Metr-la dataset, SF-PE achieves $R^2 = 0.317$, showing a 10.5% improvement over CPG-PE's 0.287. This suggests that our SF-PE is more effective at capturing long-term dependencies.

**Dataset-specific characteristic analysis:**

- **Solar dataset:** The strong periodic patterns in this dataset appear well-suited for spatiotemporal PE, as all models achieved their highest performance on this task.

Table 2: Performance comparison on six text classification tasks using the Spikformer backbone. The best results are shown in **bold**. PE types: A = absolute, F = Fused (absolute + relative). Metrics: F1 score for MRPC, Pearson Correlation for STS-B and accuracy for all other tasks. All results are averaged across 3 random seeds.

| Model | PE Type | Param(M) | Sentiment Analysis | | | | Similarity | | Inference | Avg. |
|---|---|---|---|---|---|---|---|---|---|---|
| | | | MR | SST-2 | Subj | SST-5 | MRPC | STS-B | RTE | |
| Fine-tuned BERT (Upper bound) | A | 109.8 | 86.39 | 92.01 | 95.43 | 49.87 | 89.75 | 86.47 | 69.42 | 81.33 |
| Conv-PE Zhou et al. (2022) | A | 109.8 | 71.84 | 80.17 | 88.35 | 38.69 | 68.38 | 18.71 | 52.71 | 59.84 |
| CPG-PE Lv et al. (2024) | A | 110.4 | 72.73 | 81.77 | 88.97 | 39.15 | 70.10 | 18.71 | 52.71 | 60.59 |
| SF-PE (Ours) | F | 110.4 | **73.57** | **81.83** | **89.70** | **40.05** | **70.59** | **19.24** | 52.71 | **61.10** |

- **Electricity dataset:** Despite high dimensionality (321 customers), SF-PE shows particularly strong results, indicating that our fused PE approach can effectively capture complex multivariate relationships.

- **Traffic data (Metr-la, Pems-bay):** SF-PE consistently maintains its performance advantage on the more volatile and inherently challenging traffic datasets, even with lower absolute scores.

**Comparison with upper bound:** Vanilla transformer used in SNNs is considered the performance upper bound, as the binarization process in spiking models can cause information loss compared to the continuous values used in standard transformers. Notably, in some instances, the addition of SF-PE enables spiking models to outperform the upper bound.

## 5.2 TEXT CLASSIFICATION

Table 2 presents the performance on six text classification tasks.

**Improvement in sentiment analysis:** Our SF-PE consistently surpasses both the CPG-PE baseline and the model without PE across all sentiment analysis tasks. Specifically, it achieves 73.57% accuracy on the MR task, an improvement of 0.84% over CPG-PE 72.73%. It also attains the highest performance of 40.05% in the fine-grained sentiment classification of SST-5.

**Improvement in other tasks:** On the MRPC, SF-PE improves the F1 score by 0.49% to 70.59%, compared to CPG-PE. However, for the RTE, neither CPG-PE nor SF-PE provides a performance benefit over the Spikformer baseline. This phenomenon arises because the subtle differences between sentence pairs in NLP tasks often cause the model to fall into local minima or, in the worst case, fail to converge (Lv et al., 2023). Nevertheless, while CPG-PE showed no performance improvement on the STS-B task where similar issues have been reported, SF-PE achieved a 0.53% performance gain.

**Comparison with upper bound:** While the BERT model provides upper bounds, SF-PE demonstrates substantial performance. Particularly, on the Subj (subjectivity classification) task, our method achieves 89.70%, narrowing the performance gap to BERT's 95.43%.

## 5.3 LENGTH EXTRAPOLATION ANALYSIS

Table 3: Length extrapolation evaluation on four time series forecasting tasks. Models were trained on short sequences (i.e., $L = 12$) and tested on significantly longer sequences (i.e., $L = 168$). Metrics: a higher $R^2$ indicates better performance. All results are averaged across 3 random seeds.

| Models | PE Type | Metric | Metr-la (L = 12 ->168) | | | | Pems-bay (L = 12 ->168) | | | | Solar (L = 12 ->168) | | | | Electricity (L = 12 ->168) | | | | Avg. |
|---|---|---|---|---|---|---|---|---|---|---|---|---|---|---|---|---|---|---|---|
| | | | 6 | 24 | 48 | 96 | 6 | 24 | 48 | 96 | 6 | 24 | 48 | 96 | 6 | 24 | 48 | 96 | |
| Spikformer w/CPG-PE | A | $R^2$ ↑ | .551 | .339 | **.307** | .149 | .677 | .631 | .594 | .529 | .928 | .747 | .513 | .342 | .979 | .975 | .967 | .961 | .637 |
| | | RSE↓ | .708 | .859 | **.879** | .974 | .595 | .636 | .675 | .739 | .273 | .512 | .741 | .918 | .266 | .284 | .321 | .344 | .608 |
| Spikformer w/SF-PE (Ours) | F | $R^2$ ↑ | **.601** | **.387** | .257 | **.187** | **.694** | **.679** | **.653** | **.647** | **.936** | **.764** | **.528** | **.371** | **.980** | **.977** | **.971** | **.966** | **.662** |
| | | RSE↓ | **.667** | **.827** | .910 | **.952** | **.579** | **.593** | **.617** | **.622** | **.256** | **.493** | .756 | .889 | **.263** | **.279** | **.302** | **.324** | **.583** |

We evaluate the sensitivity of our proposed method to sequence length variations by training models on short sequences ($L = 12$) and testing them on long sequences ($L = 168$). This test assesses how well the models maintain performance under extremely extrapolated conditions.

The results in Table 3 show that our SF-PE consistently achieves higher performance than CPG-PE in all cases, with an average $R^2$ of 0.662 compared to CPG-PE's 0.637. This indicates that the relative positional information within SF-PE enables the model to effectively generalize positional relationships and maintain stable prediction performance, even when the sequence length changes dramatically.

### 5.4 ABLATION STUDY

#### 5.4.1 1D VS. 2D IN SPIKING-RoPE

We proposed Spiking-RoPE, which independently encodes positional information along the temporal $t$ and spatial $l$ axes. Tab. 4 validates this design by comparing the performance of 1D and 2D in Spiking-RoPE on the Electricity dataset.

The results show a clear progression in performance. While both 1D Spiking-RoPE variants show strong performance, 2D Spiking-RoPE further improves the average $R^2$ score to 0.971. The complete SF-PE model ultimately attains the highest score of 0.973. This demonstrates a clear synergy between separating spatiotemporal features and fusing absolute with relative positional information.

Table 4: 1D vs. 2D SNN RoPE performance on Electricity dataset using Spikformer backbone. The best results are shown in **bold** and the second highest results are underlined. Metrics: a higher $R^2$ indicates better performance.

| Models | PE Type | Electricity | | | | Avg. |
|---|---|---|---|---|---|---|
| | | 6 | 24 | 48 | 96 | |
| Conv-PE | A | .959 | .955 | .955 | .954 | .956 |
| CPG-PE | A | .972 | .970 | .966 | .960 | .967 |
| 1D-Spatial RoPE | R | .975 | .972 | .966 | .960 | .968 |
| 1D-Temporal RoPE | R | .976 | .973 | .968 | .962 | .970 |
| 2D-RoPE | R | .978 | .973 | .969 | .963 | .971 |
| Spiking Fused-PE | F | **.981** | **.975** | **.972** | **.965** | **.973** |

### 5.5 RESULTS DISCUSSION AND ANALYSIS

The experimental results demonstrate the effective design of SF-PE from multiple angles:

**Empirical validation of theoretical predictions:** The phase preservation theory presented in Section 4.2.3 has been confirmed in actual experiments. Despite nonlinear transformations of LIF neurons, the consistently improved performance of models with Spiking-RoPE suggests that phase kernel preservation under linear approximation is indeed effective.

**Synergistic effect of Fused PE:** The combination of absolute PE (CPG-PE) and relative PE (Spiking-RoPE) creates synergy beyond simple performance summation. As shown in Eq. 16, this is because absolute information (amplitudes $A, B$) and relative information (trigonometric kernels $\cos\Delta, \sin\Delta$) work complementarily to expand the representation space.

**Task-specific adaptability:**

- **Time series forecasting:** The spatiotemporal separation approach of Spiking-RoPE is particularly effective for tasks where periodic patterns and long-term dependencies are important.

- **Text classification:** Relative positional information contributes to performance improvement even in natural language tasks where contextual understanding is crucial.

- **Length extrapolation:** Shows stable performance even on inputs longer than training sequences, confirming generalization ability.

## 6 CONCLUSION

We presented **Spiking Fused-PE (SF-PE)**, a spiking-friendly positional encoding that fuses absolute CPG codes with pre-spike rotary phases. Built on Spiking-RoPE and its 2D extension, our design preserves relative phase kernels under LIF dynamics while injecting complementary absolute information. Across time series and text tasks on Spikformer, SDT-V1, and QKFormer backbones, SF-PE delivers consistent accuracy gains and stronger length extrapolation without an increase in model parameters. These results validate that absolute and relative encodings are synergistic in spiking transformers and provide a principled approach for spatiotemporal PE under spiking constraints.

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

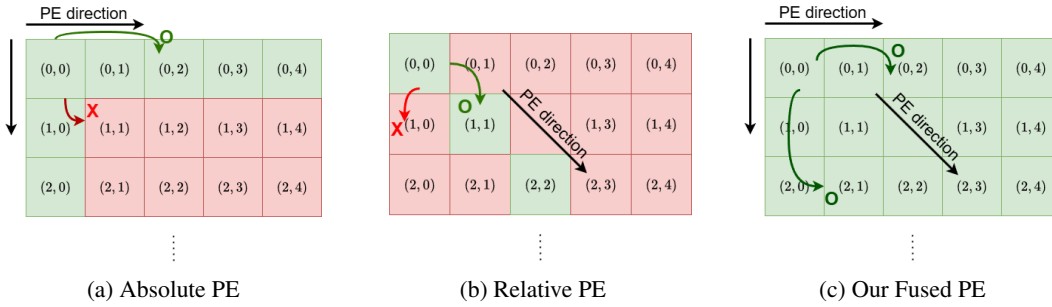

|              |              |              |
| :----------: | :----------: | :----------: |
| (a) Absolute PE | (b) Relative PE | (c) Our Fused PE |

Figure 2: Attention map activation differences between Absolute/Relative/Fused PE approaches. Green regions indicate higher activation, while red regions show lower activation. The fused PE approach demonstrates more balanced and distributed attention patterns, combining the structured patterns of absolute PE with the relative position awareness.

## A PROOF OF RELATIVE PHASE KERNEL PRESERVATION UNDER LIF DYNAMICS

This appendix provides the detailed mathematical proof for **Gap 1 Resolution (C1)** - our theoretical foundation for spiking-friendly positional encoding introduced in Section 4. These proofs establish why Spiking-RoPE succeeds where previous approaches fail, directly addressing the theoretical void identified in our gap analysis (Section 1). The results presented here underpin the practical performance gains demonstrated in Section 5.

**Theorem 1** (Relative Phase Kernel Preservation). *Let $\sigma : \mathbb{R} \to [0,1]$ be the spike probability function of a LIF neuron, and assume that batch normalization ensures $\mathbb{E}[\tilde{q}_{id}] = \mathbb{E}[\tilde{k}_{jd}] = 0$ for all dimensions $d$. Under the assumption that $\sigma$ is continuously differentiable in a neighborhood of 0 and the pre-spike activations are concentrated near zero, the expected attention score between spiked query and key vectors preserves the relative phase kernel structure:*

$$\mathbb{E}[q_i^\top k_j] = C + \alpha\langle\tilde{q}_i, \tilde{k}_j\rangle + \mathcal{O}(\epsilon^2),$$

*where $C = D\cdot\sigma(0)^2$ is a position-independent constant, $\alpha = \sigma'(0)^2 > 0$ is the sensitivity-dependent scaling factor, and $\epsilon$ quantifies the deviation from the linearization point.*

*Proof.* We proceed through several lemmas to establish the main result.

**Lemma 1** (Query/Key Setup). *According to Section 3.2 and Section 4.2, the pre-spike query and key vectors with injected positional information are given by:*

$$\tilde{q}_i = R_i q_i^{(c)}, \quad \tilde{k}_j = R_j k_j^{(c)}, \tag{19}$$

*where $R_i, R_j$ are position-dependent rotation matrices and $q_i^{(c)}, k_j^{(c)}$ are the continuous embeddings before spiking.*

**Lemma 2** (Expected Inner Product Decomposition). *Let $q_i$ and $k_j$ denote the spiked query and key vectors, where each component $q_{id}, k_{jd} \in \{0, 1\}$ is a binary random variable. Then by linearity of expectation:*

$$\mathbb{E}[q_i^\top k_j] = \mathbb{E}\left[\sum_{d=1}^{D} q_{id}k_{jd}\right] = \sum_{d=1}^{D} \mathbb{E}[q_{id}k_{jd}] \tag{20}$$

*Proof of Lemma 2.* This follows directly from the linearity of expectation operator over finite sums. □

**Lemma 3** (Binary Variable Product Expectation). *For binary random variables $q_{id}, k_{jd} \in \{0, 1\}$, the expectation of their product equals the joint probability:*

$$\mathbb{E}[q_{id}k_{jd}] = P(q_{id} = 1, k_{jd} = 1) \tag{21}$$

*Proof of Lemma 3.*

$$\mathbb{E}[q_{id}k_{jd}] = \sum_{q_{id},k_{jd}\in\{0,1\}} q_{id}k_{jd}P(q_{id},k_{jd}) \tag{22}$$

$$= 1\cdot 1\cdot P(q_{id}=1,k_{jd}=1) + \text{(other terms are zero)} \tag{23}$$

$$= P(q_{id}=1,k_{jd}=1) \tag{24}$$

$\square$

**Lemma 4** (Taylor Approximation of Spike Probability). *Let $\sigma(u) = P(s=1|u)$ be the spike probability function for input current $u$, which is monotonically increasing and continuously differentiable. Given that the SSA block applies batch normalization to ensure zero mean and unit variance (excluding bias), and assuming that the input distribution is concentrated in a neighborhood of zero, we have:*

$$\sigma(u) = \sigma(0) + \sigma'(0)u + \mathcal{O}(u^2), \tag{25}$$

*where $\sigma(0)$ represents the baseline firing probability and $\sigma'(0) > 0$ represents the sensitivity at the linearization point.*

**Lemma 5** (Taylor Approximation of Spike Probability with Remainder Bound). *Let $\sigma(u) = \Pr(s = 1 \mid u)$ be continuously differentiable in a neighborhood of $0$ and suppose $\mathbb{E}[\tilde{q}_{id}] = \mathbb{E}[\tilde{k}_{jd}] = 0$ (post-BN centering; zero-bias linear maps). Assume moreover that $\sigma$ has bounded second derivative on $[-\varepsilon,\varepsilon]$: $\sup_{|u|\le\varepsilon} |\sigma''(u)| \le M_2$. Then for $|u|,|v| \le \varepsilon$,*

$$\sigma(u)\sigma(v) = \sigma(0)^2 + \sigma'(0)^2\,uv + \sigma(0)\sigma'(0)(u+v) + \tfrac{1}{2}\sigma(0)\sigma''(0)(u^2+v^2) + R_3,$$

*and with $\mathbb{E}[u] = \mathbb{E}[v] = 0$,*

$$\left|\mathbb{E}\big[\sigma(u)\sigma(v)\big] - \big(\sigma(0)^2 + \sigma'(0)^2\mathbb{E}[uv]\big)\right| \le \tfrac{1}{2}|\sigma(0)\sigma''(0)|\big(\mathbb{E}[u^2]+\mathbb{E}[v^2]\big) + \mathbb{E}[|R_3|].$$

*Proof of Lemma 4.* By Taylor's theorem, for a continuously differentiable function $\sigma$ in a neighborhood of $u=0$:

$$\sigma(u) = \sigma(0) + \sigma'(\xi)u$$

for some $\xi$ between $0$ and $u$. Under the assumption that $\sigma$ has a bounded second derivative and the inputs are concentrated near zero, we can approximate $\sigma'(\xi) \approx \sigma'(0)$ with error $\mathcal{O}(u)$, yielding the stated result.

$\square$

**Lemma 6** (Query/Key Joint Firing Probability Approximation). *Let the joint firing probability of query and key be $P(q_{id}=1,k_{jd}=1)$. Given inputs $\tilde{q}_{id}$ and $\tilde{k}_{jd}$, instead of assuming conditional independence of spiking events, we approximate the expected joint firing rate by considering the statistical properties of the inputs.*

$$P(q_{id}=1,k_{jd}=1) = \mathbb{E}[\sigma(\tilde{q}_{id})\sigma(\tilde{k}_{jd})] \tag{26}$$

$$\approx \mathbb{E}[(\sigma(0)+\sigma'(0)\tilde{q}_{id})(\sigma(0)+\sigma'(0)\tilde{k}_{jd})] \quad \textit{(by Eq. 25)} \tag{27}$$

$$= \mathbb{E}[\sigma(0)^2 + \sigma(0)\sigma'(0)(\tilde{q}_{id}+\tilde{k}_{jd}) + \sigma'(0)^2\tilde{q}_{id}\tilde{k}_{jd}] \tag{28}$$

$$= \sigma(0)^2 + \sigma(0)\sigma'(0)(\mathbb{E}[\tilde{q}_{id}]+\mathbb{E}[\tilde{k}_{jd}]) + \sigma'(0)^2\mathbb{E}[\tilde{q}_{id}\tilde{k}_{jd}]. \tag{29}$$

*Since BN ensures $\mathbb{E}[\tilde{q}_{id}] \approx 0$ and $\mathbb{E}[\tilde{k}_{jd}] \approx 0$, the middle term vanishes. For a given token pair, the pre-spike values $\tilde{q}_{id}$ and $\tilde{k}_{jd}$ are deterministic. The expectation is taken over the stochastic spiking events, which are functions of these pre-spike values.*

$$P(q_{id}=1,k_{jd}=1) \approx \sigma(0)^2 + \sigma'(0)^2\tilde{q}_{id}\tilde{k}_{jd}. \tag{30}$$

*Proof of Lemma 6.* This follows from the derivation above using Taylor expansion and the zero-mean property ensured by batch normalization. $\square$

Now we combine all lemmas to complete the proof of Theorem 1.

Substituting the result from Lemma 6 into the expectation formula from Lemma 2:

$$\mathbb{E}[q_i^\top k_j] = \sum_{d=1}^{D} P(q_{id} = 1, k_{jd} = 1) \quad \text{(Lemma 2)} \tag{31}$$

$$= \sum_{d=1}^{D} (\sigma(0)^2 + \sigma'(0)^2 \tilde{q}_{id}\tilde{k}_{jd}) \quad \text{(Lemma 6)} \tag{32}$$

$$= D \cdot \sigma(0)^2 + \sigma'(0)^2 \sum_{d=1}^{D} \tilde{q}_{id}\tilde{k}_{jd} \tag{33}$$

$$= C + \alpha\langle \tilde{q}_i, \tilde{k}_j \rangle + \mathcal{O}(\epsilon^2) \tag{34}$$

where $C = D \cdot \sigma(0)^2$ is a position-independent constant representing the baseline co-firing expectation determined solely by the neuron's intrinsic firing rate, and $\alpha = \sigma'(0)^2 > 0$ is a positive scaling constant depending on the neuron's sensitivity.

The constant $C$ adds a uniform positive offset to the attention score expectations for all position pairs $(i, j)$. After normalizing out this constant bias effect, the attention score expectation becomes proportional to the pre-spike inner product:

$$\mathbb{E}[q_i^\top k_j] - C = \alpha\langle \tilde{q}_i, \tilde{k}_j \rangle + \mathcal{O}(\epsilon^2) \tag{35}$$

This establishes that the expected attention scores preserve the relative phase kernel structure encoded in the pre-spike embeddings, completing the proof. □

**Theorem 2** (RoPE Kernel Preservation). *When Spiking-RoPE is applied to the pre-spike embeddings such that $\tilde{q}_i = R_{\varphi(i)}q_i^{(c)}$ and $\tilde{k}_j = R_{\varphi(j)}k_j^{(c)}$, the expected attention score preserves the relative phase dependence:*

$$\mathbb{E}[q_i^\top k_j] \approx C + \alpha\big[\langle q_i^{(c)}, k_j^{(c)} \rangle \cos(\Delta_{ij}) + \langle q_i^{(c)}, Jk_j^{(c)} \rangle \sin(\Delta_{ij})\big]$$

*where $\Delta_{ij} = \varphi(i) - \varphi(j)$ is the relative phase difference and $J$ is the $90$ rotation operator.*

*Proof of Theorem 2.* This follows directly from Theorem 1 and the trigonometric expansion of the RoPE inner product $\langle \tilde{q}_i, \tilde{k}_j \rangle$ as shown in the main text. □

## B  FORMAL ANALYSIS OF APPROXIMATION ERROR BOUND

To rigorously validate the linear approximation assumption of the LIF firing function used in Theorem 1, we conduct a formal analysis of the expected approximation error. We define the total expected error $\mathcal{E}$ as the integral of the pointwise error weighted by the input probability density

$$\mathcal{E} = \mathbb{E}_{u \sim p(u)}[|\sigma(u) - \hat{\sigma}(u)|] = \int_{-\infty}^{\infty} e(u)p(u)\,du, \tag{36}$$

where $u$ is the pre-spike membrane potential, $\sigma(u)$ is the actual surrogate gradient function (ATan), $\hat{\sigma}(u)$ is the linear approximation, and $p(u)$ is the probability density function of $u$. Based on the empirical observations in Figure 3 and Figure 4, we decompose this integral into a **Core Linear Region** ($|u| \leq 1$) and a **Tail Region** ($|u| > 1$)

$$\mathcal{E} = \underbrace{\int_{|u| \leq 1} e(u)p(u)\,du}_{\text{Term 1: Core Linear Region}} + \underbrace{\int_{|u| > 1} e(u)p(u)\,du}_{\text{Term 2: Tail Region}}. \tag{37}$$

### B.1 EMPIRICAL VALIDATION OF INPUT DISTRIBUTION

We analyze the distribution of pre-spike membrane potentials $p(u)$ in the trained model. As shown in Figure 3, the use of batch normalization effectively stabilizes the distribution of Query (Q) and Key (K) potentials.

- **Zero-Centered & Narrow:** The distributions are consistently centered near zero ($\mu \approx 0$) with a standard deviation of $\sigma \approx 1.1$ across all training epochs.
- **RoPE Invariance:** The distributions before and after applying RoPE (blue vs. red histograms in Figure 3) overlap almost perfectly, indicating that the rotation operation does not distort the statistical properties of the input, preserving the validity of the approximation throughout the network depth.

### B.2 ERROR BOUND DECOMPOSITION

Combining the distribution analysis with the linearity analysis in Figure 4, we evaluate the two terms of the expected error.

1) **Term 1: Core Linear Region** ($|u| \leq 1$)
   Figure 4 demonstrates that the interval $[-1, 1]$ corresponds to the inflection point of the ATan surrogate function. In this region, the linear approximation (green dashed line) and the actual function (red solid line) are nearly indistinguishable.

   - **Observation:** The pointwise error $e(u)$ in this region is essentially negligible ($\approx 0$).
   - **Density:** Empirically, approximately **73%** of the total probability mass lies within this high-precision core.
   - **Result:** The contribution of Term 1 to the total error is minimal due to the vanishing $e(u)$.

2) **Term 2: Tail Region** ($|u| > 1$)
   Outside the unit interval, the pointwise error $e(u)$ begins to increase as the surrogate function saturates. However, the contribution of this error is suppressed by the vanishing probability density.

   - **Observation:** As shown in Figure 3, the input density $p(u)$ decays rapidly (sub-Gaussian tail) for $|u| > 1$.
   - **Density:** Over **95%** of the data falls within the $[-2, 2]$ range. The probability mass in the extreme saturation region ($|u| > 2$), where the linear assumption would fail significantly, is statistically insignificant ($< 5\%$).
   - **Result:** The exponential decay of $p(u)$ dominates the linear growth of $e(u)$, ensuring that the integral of Term 2 remains tightly bounded.

The quantitative analysis in Figure 4 confirms a consistently high correlation ($> 0.96$) between the ATan function and its linear approximation within the effective data range. Consequently, the total expected error $\mathcal{E}$ is bounded by a small constant

$$\mathcal{E} \approx \epsilon_{core} \cdot P(|u| \leq 1) + C_{tail} \cdot P(|u| > 2) \approx 0. \tag{38}$$

This provides a strong empirical justification that the phase preservation properties derived in Theorem 1 hold in expectation under the actual operating conditions of the SNN.

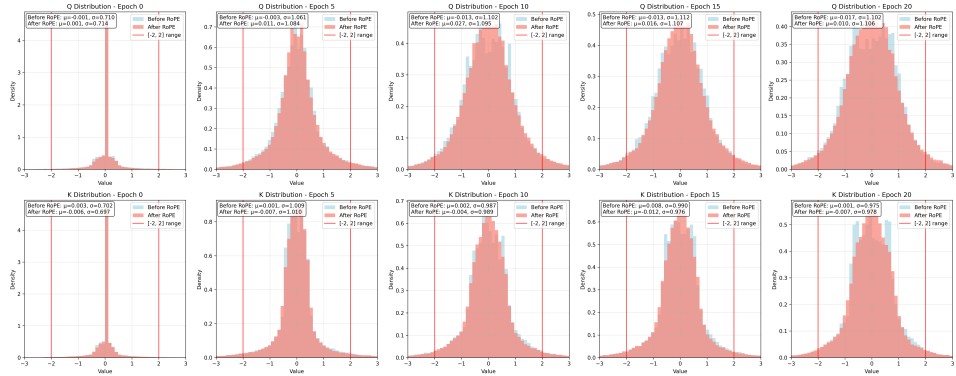

Figure 3: Pre-spike Query/Key value distribution on every five epochs (0 to 20 epoch).

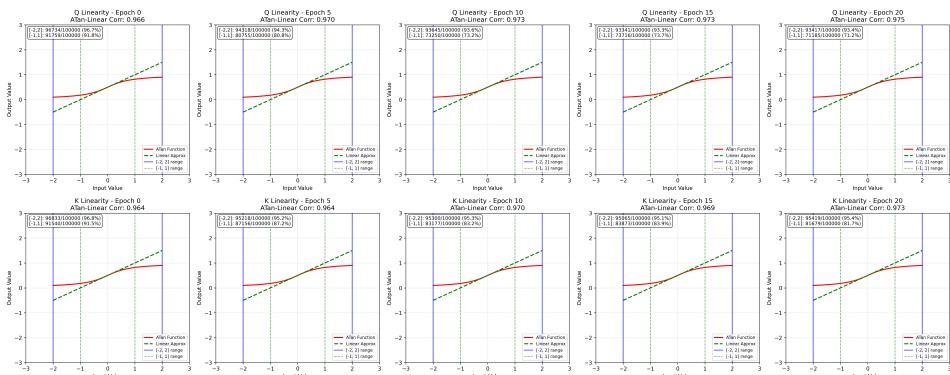

Figure 4: Comparison of the ATan surrogate function and its first-order Taylor linear approximation around zero (Lemma 4).

## C EXPERIMENTAL SETTINGS

This appendix provides comprehensive experimental setup details for all experiments reported in Section 5, including datasets, hyperparameters, and evaluation metrics.

### C.1 DATASETS

#### C.1.1 TIME SERIES FORECASTING

- **Metr-la (Jagadish et al., 2014):** Traffic speed dataset from Los Angeles County highways.
- **Pems-bay (Li et al., 2018):** Traffic speed dataset from the California Bay Area.
- **Solar (Lai et al., 2018):** Solar power generation dataset from the 2006 US National Solar Radiation Database.
- **Electricity (Lai et al., 2018):** Power consumption dataset recording electricity usage of 321 customers.

#### C.1.2 TEXT CLASSIFICATION

- **MR (Pang & Lee, 2005):** Sentiment analysis dataset for classifying positive/negative movie reviews.
- **SST-2 (Socher et al., 2013):** Binary sentiment classification dataset for movie review sentences.
- **SST-5 (Socher et al., 2013):** Sentiment classification dataset with five fine-grained classes (very positive to very negative).

- **Subj (Conneau & Kiela, 2018):** Dataset for classifying sentences as subjective or objective.
- **MRPC (Dolan & Brockett, 2005):** Dataset for determining whether two sentences are semantically equivalent.
- **STS-B (Cer et al., 2017): Dataset for sentence pairs drawn from news headlines, video and image captions.**
- **RTE (Haim et al., 2006):** Dataset for recognizing logical entailment relationships between two sentences.

## C.2 HYPERPARAMETER SETTINGS

All experiments are repeated three times under the same random seed, and we report the average performance. Detailed hyperparameters for each task are as follows.

### C.2.1 COMMON PARAMETERS

LIF neuron parameters commonly used in all experiments are shown in Table 5.

Table 5: Experiment configuration and hyperparameter settings for time series forecasting and text classification tasks, including LIF neuron parameters, training settings, and positional encoding specific parameters.

| Parameter | Time Series Forecasting | Text Classification |
|---|---|---|
| *Common LIF Neuron Parameters* | | |
| Membrane leak time constant ($\tau$) | 2.0 | |
| Common firing threshold ($U_{thr}$) | 0.8 | |
| *Training & Model Parameters* | | |
| Batch Size | 64 | 64 |
| Learning Rate | 0.001 (1e-3) | 5e-4 |
| Optimizer | Adam | Adam |
| Embedding Dimension | 256 | 768 |
| Layer Depth | 2 | 12 |
| Time Steps ($T$) | 4 | 4 |
| Attention Heads | 8 | 8 |
| Max Sequence Length | - | 256 |
| *Positional Encoding Parameters* | | |
| CPG Neurons ($N_{CPG}$) | 40 | 20 |
| Spiking-RoPE Base ($B$) | 10000.0 | 10000.0 |

## D SPIKING-ROPE IMPLEMENTATION

This appendix provides detailed implementation details for the Spiking-RoPE method introduced in Section 4.2, with specific focus on the tensor operations and computational procedures referenced in the main methodology.

This section describes how to implement Spiking-RoPE from a tensor dimension perspective, referencing the tensor transformation of the overall process as shown in Fig. 5. The core of Spiking-RoPE is to apply positional information in rotational form to query ($Q$) and key ($K$) tensors with continuous values before binarization.

### D.1 1D IMPLEMENTATION IN SPIKING-ROPE

1D Spiking-RoPE encodes relative positional information along the sequence length axis $L$ or time axis $T$. Here, we focus our explanation on the sequence length axis.

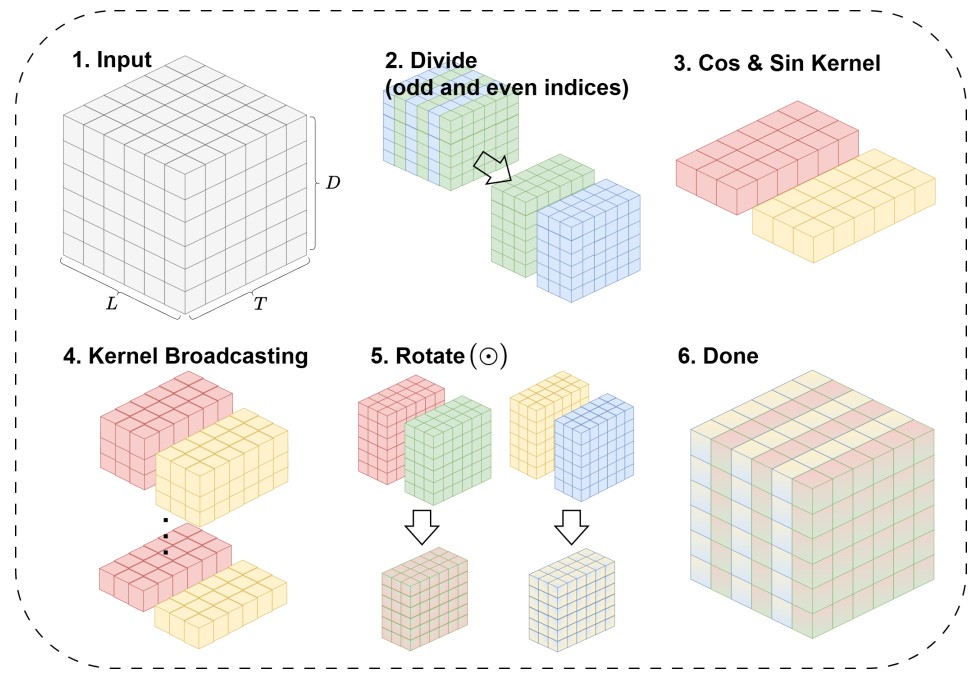

Figure 5: Spiking-RoPE tensor transformation.

1) **Input Tensor and Parameter Definition**

   - **Input Tensor**: A Query or Key tensor $X \in \mathbb{R}^{T \times L \times D}$ is given, where $T$ represents time steps, $L$ represents sequence length, and $D$ represents feature dimension.
   - **Position and Frequency Indices**:
     - Position index: $m \in \{0, 1, ..., L-1\}$
     - Frequency index: $i \in \{0, 1, ..., D/2 - 1\}$
   - **Rotation Frequency**: Calculate $\theta_i = B^{-2i/D}$ corresponding to each frequency index, where $B$ is a hyperparameter (e.g., 10000).

2) **Channel Separation (Divide)**

   - Separate the input tensor $X$ into even-indexed channels ($X_{\text{even}}$) and odd-indexed channels ($X_{\text{odd}}$) based on the last dimension $D$.
   - $X_{\text{even}}, X_{\text{odd}} \in \mathbb{R}^{T \times L \times (D/2)}$

3) **Rotation Kernel Generation (Cos & Sin Kernel)**

   - Generate Cosine and Sine kernel tensors by combining position $m$ and frequency index $i$ (Eq. 8).
   - `cos_kernel` $\in \mathbb{R}^{L \times D/2}$: Each element has the value $\cos(m\theta_i)$.
   - `sin_kernel` $\in \mathbb{R}^{L \times D/2}$: Each element has the value $\sin(m\theta_i)$.

4) **Rotation Application (Rotate)**

   - Apply the generated kernels to the separated channels through broadcasting. `cos_kernel` and `sin_kernel` are broadcasted to the $T$ dimension and operated with $X_{\text{even}}$ and $X_{\text{odd}}$.
   - Calculate the following according to RoPE's rotation formula, where the multiplication is the Hadamard product:

$$X'_{\text{even}} = X_{\text{even}} \cdot \texttt{cos\_kernel} - X_{\text{odd}} \cdot \texttt{sin\_kernel}$$

$$X'_{\text{odd}} = X_{\text{even}} \cdot \texttt{sin\_kernel} + X_{\text{odd}} \cdot \texttt{cos\_kernel}$$

   - Both $X'_{\text{even}}$ and $X'_{\text{odd}}$ have dimensions $(T, L, D/2)$.

5) **Channel Combination (Done)**

- Interleave the rotation-applied $X'_{\text{even}}$ and $X'_{\text{odd}}$ back to their original order along the last dimension $D$ to complete the final tensor $X_{\text{rotated}} \in \mathbb{R}^{T \times L \times D}$. This tensor contains relative positional information for the sequence length axis.

---

**Algorithm 1** 1D Spiking-RoPE algorithm implementation. The algorithm applies position-dependent rotations along the sequence length axis by computing trigonometric kernels and rotating even/odd channel pairs to inject relative positional information before spike binarization.

---

**Require:** $x \in \mathbb{R}^{T \times L \times D}$ (Input tensor), $B \in \mathbb{R}$ (Base value for frequency computation)
**Ensure:** $x' \in \mathbb{R}^{T \times L \times D}$ (Position-encoded tensor)
1: **function** APPLY1DSPIKINGROPE($x, B$)
2:     $(T, L, D) \leftarrow$ shape of $x$
3:     $x' \leftarrow$ copy of $x$

4:     **for all** $i \in \{0, 1, \ldots, D/2 - 1\}$ **do**
5:         $\theta_i \leftarrow 1/B^{2i/D}$                                        ▷ Compute rotation frequencies
6:     **end for**

7:     **for all** $m \in \{0, 1, \ldots, L - 1\}$ **do**              ▷ Apply rotation for each position $m$
8:         **for all** $i \in \{0, 1, \ldots, D/2 - 1\}$ **do**        ▷ For each channel pair $(2i, 2i + 1)$
9:             $c \leftarrow \cos(m\theta_i)$
10:            $s \leftarrow \sin(m\theta_i)$

11:            $x_{m,2i} \leftarrow x[:, m, 2i]$                      ▷ Get the channel vector across time T
12:            $x_{m,2i+1} \leftarrow x[:, m, 2i + 1]$

13:            $x'[:, m, 2i] \leftarrow x_{m,2i} \cdot c - x_{m,2i+1} \cdot s$           ▷ Apply 2D rotation
14:            $x'[:, m, 2i + 1] \leftarrow x_{m,2i} \cdot s + x_{m,2i+1} \cdot c$
15:         **end for**
16:     **end for**

17:     **return** $x'$
18: **end function**

---

## D.2 2D IMPLEMENTATION IN SPIKING-ROPE

2D Spiking-RoPE considers the spatiotemporal characteristics of SNNs and encodes positional information for the sequence length axis $L$ and time axis $T$ independently.

1) **Feature Dimension Partitioning**
   - Divide the last feature dimension $D$ of the input tensor $X \in \mathbb{R}^{T \times L \times D}$ into two equal-sized blocks.
     - $X_l \in \mathbb{R}^{T \times L \times (D/2)}$: Part to be used for sequence length ($L$) axis rotation
     - $X_t \in \mathbb{R}^{T \times L \times (D/2)}$: Part to be used for time ($T$) axis rotation
2) **Independent Application of 1D RoPE to Each Axis**
   - **(a) Sequence Length Axis Rotation**:
     - Apply the 1D Spiking-RoPE process described in Appendix D.1 directly to tensor $X_l$.
     - Here, the feature dimension becomes $D/2$, and the rotation kernel is generated based on sequence length $L$.
     - This yields $X'_l \in \mathbb{R}^{T \times L \times (D/2)}$.
   - **(b) Time Axis Rotation**:
     - Apply the same 1D Spiking-RoPE process to tensor $X_t$, but change the positional reference to the time axis $T$.

- **Rotation Kernel Generation**: Use position index $t \in \{0, 1, ..., T-1\}$ to generate `cos_kernel_t` and `sin_kernel_t` of size $(T, D/4)$ (since feature dimension $D/2$ is again divided into even/odd, resulting in $D/4$).
- **Rotation Application**: Broadcast the generated time kernels to the $L$ dimension (sequence length axis) and apply to $X_t$.
- This yields $X_t' \in \mathbb{R}^{T \times L \times (D/2)}$.

3) **Final Tensor Combination**

- Concatenate the tensors $X_l'$ and $X_t'$, which have been independently rotated for both axes, along the last feature dimension.
- Finally obtain $X_{\text{2D\_rotated}} = \text{concat}([X_l', X_t']) \in \mathbb{R}^{T \times L \times D}$ with both spatiotemporal positional information encoded.

---

**Algorithm 2** 2D Spiking-RoPE algorithm implementation. This algorithm extends 1D Spiking-RoPE to handle spatiotemporal data by independently applying rotations along both sequence length and time axes, with feature dimensions partitioned equally between the two axes.

**Require:** $x \in \mathbb{R}^{T \times L \times D}$ (Spatiotemporal input tensor), $B \in \mathbb{R}$ (Base value for frequency computation)
**Ensure:** $x' \in \mathbb{R}^{T \times L \times D}$ (Spatiotemporally encoded tensor)
1: **function** APPLY2DSPIKINGROPE($x, B$)
2:     $x_L \leftarrow x[:, :, : D/2]$                       ▷ Partition features for length (L) axis
3:     $x_T \leftarrow x[:, :, D/2 :]$                      ▷ Partition features for time (T) axis

4:     $x_L' \leftarrow \text{Apply1DSpikingRoPE}(x_L, B)$           ▷ Encode the length axis
                                    ▷ Encode the time axis by treating T as the sequence dimension
5:     $x_T^{\text{perm}} \leftarrow \text{Transpose}(x_T, \text{axes} = (0, 1))$
6:     $x_T'^{\text{perm}} \leftarrow \text{Apply1DSpikingRoPE}(x_T^{\text{perm}}, B)$
7:     $x_T' \leftarrow \text{Transpose}(x_T'^{\text{perm}}, \text{axes} = (0, 1))$

8:     $x' \leftarrow \text{Concatenate}(x_L', x_T', \text{axis} = 2)$         ▷ Combine the encoded partitions

9:     **return** $x'$
10: **end function**

---

# E   COMPARISON WITH GRAY/LOG PE

While Gray-PE and Log-PE represent advanced relative positional encoding methods for SNNs, their official code has not been fully released at the time of this writing. To ensure reproducibility, we limit our direct comparison to the publicly available Spikformer backbone, for which we utilize Gray-PE and Log-PE results based on their paper. This allows for a fair and direct performance evaluation against our proposed SF-PE.

Table 6: Performance comparison on time series forecasting across 4 benchmark datasets with prediction lengths of 6, 24, 48, and 96 hours. Best results are highlighted in **bold**. PE types: R = relative, F = Fused (absolute + relative). Metrics: higher $R^2$ and lower RSE indicate better performance. All results are averaged across 3 random seeds.

| Models | PE Type | Metric | Metr-la (L = 12) | | | | Pems-bay (L = 12) | | | | Solar (L = 168) | | | | Electricity (L = 168) | | | | Avg. |
|---|---|---|---|---|---|---|---|---|---|---|---|---|---|---|---|---|---|---|---|
| | | | 6 | 24 | 48 | 96 | 6 | 24 | 48 | 96 | 6 | 24 | 48 | 96 | 6 | 24 | 48 | 96 | |
| Spikformer w/Gray-PE Lv et al. (2025) | R | $R^2 \uparrow$ | .728 | .544 | .414 | .295 | .782 | **.724** | .694 | **.673** | .936 | .840 | .756 | .710 | .974 | .972 | .966 | .962 | .748 |
| | | RSE↓ | .546 | .706 | .806 | .885 | .506 | .578 | .597 | .618 | .257 | .409 | .507 | .546 | .276 | .304 | .320 | .342 | .513 |
| Spikformer w/Log-PE Lv et al. (2025) | R | $R^2 \uparrow$ | .735 | .535 | .424 | .290 | **.789** | .717 | .691 | .670 | .933 | .841 | .758 | .734 | .978 | .974 | .968 | .964 | .750 |
| | | RSE↓ | .543 | .719 | .799 | .876 | **.496** | **.575** | .601 | .620 | .265 | .408 | .504 | .525 | .272 | .300 | .314 | .340 | .509 |
| Spikformer w/SF-PE (Ours) | F | $R^2 \uparrow$ | **.739** | **.561** | **.432** | **.317** | .783 | .713 | **.698** | .670 | **.939** | **.877** | **.782** | **.752** | **.981** | **.975** | **.972** | **.965** | **.760** |
| | | RSE↓ | **.538** | **.698** | **.795** | **.871** | .499 | .576 | **.593** | .618 | **.251** | **.362** | **.479** | **.511** | **.240** | **.280** | **.300** | **.336** | **.497** |

The experimental results presented in Table 6 clearly demonstrate the superiority of SF-PE. Across all four time series forecasting benchmarks, our model consistently outperforms both Gray-PE and

Log-PE. On average, SF-PE achieves a higher $R^2$ score 0.760 compared to both Gray-PE and Log-PE. While the model demonstrated superior or competitive results across most benchmarks, particularly in long-term forecasting on the Metr-la and Solar datasets, its performance on the Pems-bay dataset was comparable to the other relative PE methods. This overall strong performance validates that the synergistic fusion of absolute and relative positional encodings in our model provides a robust advantage over methods that use relative encoding alone.

# F   ROTATION FREQUENCY VARIATION

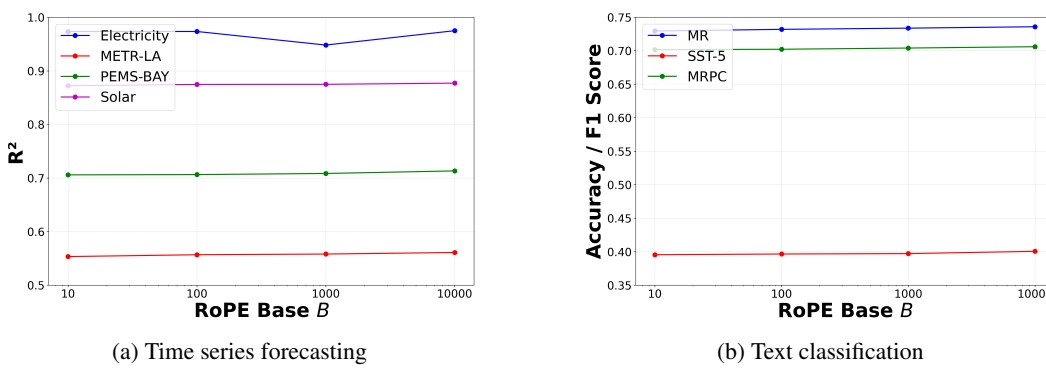

(a) Time series forecasting                      (b) Text classification

Figure 6: Performance versus RoPE base parameter $B$.

The base value $B$ is a core hyperparameter of Spiking-RoPE, which determines the rotation frequency. Appropriate frequency selection enables the model to effectively distinguish relative distances of various lengths. Too high frequencies may focus excessively on short-range relationships and miss long-range dependencies, while too low frequencies make fine positional distinctions difficult.

Our experiments, illustrated in Figure 6, demonstrate that the model is robust to the parameter $B$, with performance consistently peaking at $B = 10000$ across all tested datasets. We therefore adopt this value as the default for $B$ in all our experiments.

# G   SPACE AND TIME EFFICIENCY COMPARISON

Table 7: Efficiency comparison on the Electricity task using Spikformer

|  | Params(M) | GFLOPs | Avg training time per epoch(sec) | Avg inference time per epoch(sec) |
|---|---|---|---|---|
| Spikformer w/Conv-PE Zhou et al. (2022) | 1.67 | 72.07 | 42.56 | 5.16 |
| Spikformer w/CPG-PE Lv et al. (2024) | 1.7 | 72.08 | 42.67 | 5.23 |
| Spikformer w/SF-PE (Ours) | 1.7 | 72.09 | 42.74 | 5.26 |

To evaluate the impact of SF-PE on computational and training efficiency compared to existing methods, we measured the computational costs of three different PE schemes on the Electricity task using Spikformer. The comparison metrics include the number of parameters, GFLOPs, average training time per epoch, and average inference time per epoch. We report the per-epoch average time because the total number of epochs required for full training varies across runs due to the inherent uncertainty of spikes.

Table 7 demonstrates that SF-PE incurs minimal computational overhead. The proposed method requires no additional parameters, resulting in the exact same parameter count as CPG-PE and a negligible difference in GFLOPs. Furthermore, the time cost increase over CPG-PE is trivial, and the computational overhead remains within approximately 0.2 seconds compared to Spikformer.

## H    PRE-SPIKE VS. POST-SPIKE FOR SF-PE

Table 8: Pre-spike and Post-spike RoPE comparison on four time series forecasting tasks.

| Models | PE Type | Metric | Metr-la (L = 12) | | | | Pems-bay (L = 12) | | | | Solar (L = 168) | | | | Electricity (L = 168) | | | | Avg. |
|---|---|---|---|---|---|---|---|---|---|---|---|---|---|---|---|---|---|---|---|
| | | | 6 | 24 | 48 | 96 | 6 | 24 | 48 | 96 | 6 | 24 | 48 | 96 | 6 | 24 | 48 | 96 | |
| Spikformer w/Conv-PE Zhou et al. (2022) | R | $R^2$ ↑ | .713 | .527 | .399 | .267 | .773 | .697 | .686 | .667 | .929 | .828 | .744 | .674 | .959 | .955 | .955 | .954 | .733 |
| | | RSE↓ | .565 | .725 | .818 | .903 | .514 | .594 | .606 | .621 | .272 | .426 | .519 | .586 | .373 | .371 | .379 | .382 | .541 |
| Spikformer w/SF-PE (Post-spike) | F | $R^2$ ↑ | .558 | .358 | .281 | .145 | .682 | .669 | .655 | .646 | .881 | .791 | .735 | .689 | .941 | .939 | .932 | .929 | .677 |
| | | RSE↓ | .702 | .846 | .895 | .976 | .591 | .602 | .615 | .623 | .355 | .470 | .529 | .573 | .431 | .439 | .454 | .367 | .592 |
| Spikformer w/SF-PE (Pre-spike) | F | $R^2$ ↑ | .739 | .561 | .432 | .317 | .783 | .713 | .698 | .670 | .939 | .877 | .782 | .752 | .981 | .975 | .972 | .965 | .760 |
| | | RSE↓ | .538 | .698 | .795 | .871 | .499 | .576 | .593 | .618 | .251 | .362 | .479 | .511 | .240 | .280 | .300 | .336 | .497 |

To validate the theoretical implementation of RoPE in SNNs, we conducted a comparative experiment between pre-spike and post-spike RoPE applications. Table 8 presents the performance comparison on the time-series forecasting task using the Spikformer backbone.

Post-spike RoPE faces two primary issues. First, RoPE operates by rotating vectors by specific angles. Applying this transformation to binary spikes generates floating-point values, thereby destroying the inherent discrete nature of SNNs. Second, applying geometric rotations to low-resolution binary spikes introduces quantization errors and information distortion. As shown in Table 8, the post-spike method yields an average $R^2$ of 0.677, showing significant performance degradation compared to the baseline Conv-PE.

In contrast, our proposed pre-spike method applies rotation to the continuous membrane potentials prior to the LIF binarization step. This approach ensures high precision in the rotation operation without information loss. Furthermore, as proved in Theorem 1, the relative phase information encoded in the membrane potential is preserved in the form of statistical expectation even after passing through the nonlinear LIF dynamics. Consequently, the pre-spike method achieved an average $R^2$ of 0.760, demonstrating consistent performance improvements over the post-spike approach. This empirically validates our theoretical assertion that the pre-spike strategy is essential for effective relative positional encoding in SNNs.

## I    LIMITATIONS

The core theoretical foundation of this study critically depends on approximating the firing probability function of LIF neurons using a first-order Taylor series for phase preservation proof. This linear approximation is most effective when the pre-spike membrane potential distribution is stabilized near zero mean by Batch Normalization and has a narrow unimodal form. However, under neuron saturation states or broad distributions that may occur in real networks, the approximation accuracy may degrade, which can weaken the phase preservation effect. Although experimental results validate the effectiveness of Spiking-RoPE and its underlying approximation, the model's behavior in extreme sparse or saturated environments with complex neuron models requires additional research. Future work should verify Spiking-RoPE 's generalizability by expanding the experimental focus from sequential data to the vision domain of images.

## THE USE OF LARGE LANGUAGE MODELS

**Tool & Version**: Gemini (Google, 2025-09)
**Research Stage**: Generated visualization scripts.
**Writing Stage**: Language editing of author-drafted text for clarity and conciseness.
**Human Oversight**: All outputs reviewed/edited by the authors; authors accept full responsibility for the content.

