# OpenReview forum: "SF-PE: A Synergistic Fusion of Absolute and Relative Positional Encoding for Spiking Transformers"
_ICLR.cc/2026/Conference — Submitted to ICLR 2026_

### Official Review · Reviewer_LdHq · 2025-10-20

**Soundness:** 3
**Presentation:** 2
**Contribution:** 2
**Rating:** 2
**Confidence:** 5

**Summary:**

The paper studies how spike binarization and LIF dynamics distort positional signals in spiking Transformers, weakening self-attention.
It introduces **Spiking-RoPE**, which applies rotary positional encoding to queries/keys *before* binarization so relative phase information is preserved under LIF.
A **2-D Spiking-RoPE** decouples sequence and time rotations to explicitly model spatiotemporal relations.
To combine absolute and relative cues, the authors fuse CPG-based absolute spikes with Spiking-RoPE into **Spiking Fused-PE (SF-PE)**, yielding complementary row/column vs. diagonal attention patterns.
A theoretical analysis supports pre-spike rotations as intrinsically compatible with LIF, and experiments across time-series forecasting and text classification on multiple spiking backbones show consistent improvements and better length extrapolation.
Comprehensive ablations on rotation bases and 1-D vs. 2-D variants corroborate the design’s robustness.

**Strengths:**

The paper redesigns RoPE specifically for SNNs by applying it pre-spike, extends it to a 2-D spatiotemporal variant, and fuses absolute CPG-PE with relative Spiking-RoPE into SF-PE to capture complementary row/column and diagonal structures. It provides a statistical-expectation analysis showing phase preservation through LIF, includes thorough ablations (1-D vs 2-D; RoPE base), and reports solid results across time-series and text on multiple spiking backbones. The method is clearly motivated, the pre-spike rotation pipeline is well articulated with equations and design rationale, and assumptions are explicitly stated.

**Weaknesses:**

1) The paper motivates Spiking-RoPE as a **pre-spike** operation (before LIF binarization) to justify “phase preservation under LIF.” In the released code (`spikformer_cpg_rope.py`, self-attention forward), Q/K pass through `self.q_lif` / `self.k_lif` **first**, and only then `apply_spiking_rotary_pos_emb(...)` is applied—i.e., RoPE is **post-spike**. Under this code path, the pre-spike theoretical claim does not hold as implemented.

2) The manuscript emphasizes 2D spatiotemporal rotation and fusing absolute CPG with Q/K inputs. The public model instantiates a 1D rotary embedding (`RotaryEmbedding1DSpatial`) and injects CPG via `CPGLinear` on the encoder path rather than at Q/K. If this is intended to be equivalent, that needs justification; otherwise it should be documented as an implementation deviation or the stated 2D variant should be released.

**Questions:**

Q1. The paper argues for **pre-spike** RoPE; the released code implements **post-spike**. Which path produced the reported results? If post-spike was used, please revise the theory accordingly; if pre-spike was used, provide the corresponding implementation and report a direct pre- vs post-spike comparison.

Q2. Current code uses `RotaryEmbedding1DSpatial` and fuses CPG via `CPGLinear` on the encoder path rather than at Q/K. (i) Are these implementations theoretically equivalent to the paper’s formulas? If not, please mark as a deviation. (ii) Release and evaluate the full **2D** variant and the **Q/K-side fusion** described in the manuscript, with side-by-side results.

---

> ### Author Response · Authors · 2025-11-21
>
> **W1, W2, Q2. Submission of incorrect code**
>
> We appreciate your valuable review. We would like to address the discrepancies found in the code uploaded in the previous submission.
>
> 1. Post-spike RoPE in SSA (`Spikformer_cpg_rope.py`, `forward` method in `SSARoPE`): The uploaded code contained the post-spike implementation used for testing purposes, rather than the pre-spike RoPE used in the paper.
> 2. RotaryEmbedding1DSpatial for RoPE (`Spikformer_cpg_rope.py`, `SSARoPE` initialization): The uploaded code was configured to use `1DSpatial` RoPE instead of the final 2D RoPE method.
>
> We believe these discrepancies may have prevented our contributions from being fully recognized. Therefore, we have restated the paper’s contributions and novelty under these corrections. Our proposed method is the first to apply RoPE in SNNs by proving that the effect of RoPE is preserved in LIF activity. Furthermore, we explicitly model the intrinsic spatiotemporal characteristics of SNNs by proposing 2D RoPE, which independently rotates the sequence and time axes. We introduce SF-PE, a Fused-PE scheme that combines this relative PE (2D RoPE) with the SOTA absolute PE method (CPG-PE), demonstrating consistent performance improvements in both time-series forecasting and NLP tasks.
>
> In particular, through this rebuttal, we have strengthened our contributions and demonstrated the effectiveness of our method more extensively by adding experiments on computational overhead, the STS-B task in NLP, extended length extrapolation, and validation of the linear approximation assumption for the spike firing function in Sec. 4.2.3.
>
> **Q1. Pre-spike vs. Post-spike for SF-PE**
>
> We have resolved the aforementioned issues, and we confirm that all experiments in the paper were conducted using pre-spike and RotaryEmbedding2D. Furthermore, we have added comparative results between pre- and post-spike implementations in Appendix H, verifying that RoPE yields valid performance improvements only in the pre-spike setting. Applying RoPE, which rotates vectors by specific angles, to the Query and Key in a post-spike state destroys the inherent discrete nature of SNNs. Moreover, applying RoPE to low-resolution binary spikes introduces severe information distortion. Indeed, our comparative experiments confirmed that the post-spike method resulted in an $R^2$ of 0.677, a significant degradation compared to the baseline performance. In contrast, our proposed pre-spike method applies rotation to the continuous membrane potentials prior to LIF binarization. This approach improves the precision of the rotation operation, and as proven in Theorem 1, the relative phase information is preserved as a statistical expectation even after passing through the nonlinearity of LIF neurons, achieving consistent performance improvements.
> |                                 |        | Metr-la (L = 12) |      |      |      | Pems-bay (L = 12) |      |      |      | Solar (L = 168) |      |      |      | Electricity (L = 168) |      |      |      |      |
> |---------------------------------|:------:|:----------------:|:----:|:----:|:----:|:-----------------:|:----:|:----:|:----:|:---------------:|:----:|:----:|:----:|:---------------------:|:----:|:----:|:----:|:----:|
> | Models                          | Metric |         6        |  24  |  48  |  96  |         6         |  24  |  48  |  96  |        6        |  24  |  48  |  96  |           6           |  24  |  48  |  96  | Avg. |
> | Spikformer w/Conv-PE            |   R^2  |       .713       | .527 | .399 | .267 |        .773       | .697 | .686 | .667 |       .929      | .828 | .744 | .674 |          .959         | .955 | .955 | .954 | .733 |
> |                                 |   RSE  |       .565       | .725 | .818 | .903 |        .514       | .594 | .606 | .621 |       .272      | .426 | .519 | .586 |          .373         | .371 | .379 | .382 | .541 |
> | Spikformer w/SF-PE (Post-spike) |   R^2  |       .558       | .358 | .281 | .145 |        .682       | .669 | .655 | .646 |       .881      | .791 | .735 | .689 |          .941         | .939 | .932 | .929 | .677 |
> |                                 |   RSE  |       .702       | .846 | .895 | .976 |        .591       | .602 | .615 | .623 |       .355      | .470 | .529 | .573 |          .431         | .439 | .454 | .367 | .592 |
> | Spikformer w/SF-PE (Pre-spike)  |   R^2  |       .739       | .561 | .432 | .317 |        .783       | .713 | .698 | .670 |       .939      | .877 | .782 | .752 |          .981         | .975 | .972 | .965 | .760 |
> |                                 |   RSE  |       .538       | .698 | .795 | .871 |        .499       | .576 | .593 | .618 |       .251      | .362 | .479 | .511 |          .240         | .280 | .300 | .336 | .497 |

---

### Official Review · Reviewer_zk6P · 2025-10-30

**Soundness:** 2
**Presentation:** 2
**Contribution:** 2
**Rating:** 4
**Confidence:** 5

**Summary:**

This paper introduces Spiking Fused-Positional Encoding (SF-PE) for spiking transformers, which integrates absolute positional encoding (CPG-PE) and relative positional encoding (Spiking-RoPE) to resolve the distortion of positional signals in spiking neural networks (SNNs)—a problem caused by spike binarization and the nonlinear dynamics of Leaky Integrate-and-Fire (LIF) neurons. The authors provide theoretical proof that Spiking-RoPE preserves relative phase kernels in statistical expectation under LIF dynamics, and validate SF-PE across two tasks (time-series forecasting and text classification) and three spiking backbones (Spikformer, SDT-V1, QKFormer). Results show consistent accuracy improvements and enhanced length extrapolation, with ablations supporting the design of 2D Spiking-RoPE and rotation bases.

**Strengths:**

1. It rigorously proves that Spiking-RoPE preserves relative phase kernels in statistical expectation under LIF dynamics, addressing the lack of theoretical analysis for positional information preservation in existing SNN transformers (Gap 1).
2. The 2D Spiking-RoPE explicitly models spatiotemporal relationships by decoupling sequence and time axes, solving the limitation of most PEs treating position as one-dimensional (Gap 3), and ablation experiments confirm its superiority over 1D variants.

**Weaknesses:**

1. I checked the code and found that the authors merely fused the RoPE code crudely into Spiking Self-Attention without analyzing the properties of spikes; theoretically, RoPE cannot work on binary matrices, so I am skeptical about the performance improvements claimed by the authors.
2. The theoretical analysis of Spiking-RoPE’s phase preservation under LIF dynamics relies on the strong assumption that the firing probability function of LIF neurons operates almost linearly, yet the paper provides no verification of the validity range or error bounds of this assumption.
3. In the text classification experiments, the paper shows that neither CPG-PE nor SF-PE improves performance on the RTE task, but it fails to analyze the reasons for this task-specific ineffectiveness (e.g., whether it is related to spatiotemporal modeling or relative position encoding).
4. The paper claims that SF-PE enhances length extrapolation capabilities, but it only mentions the analysis in Appendix D without presenting key results (e.g., performance degradation trends on sequences longer than training lengths) in the main text.

**Questions:**

See weakness.

---

> ### Author Response · Authors · 2025-11-21
>
> We appreciate your valuable review. We would like to address the issues found in the code uploaded in the previous submission.
>
> 1. Post-spike RoPE in SSA (`forward` method in `SSARoPE`): The uploaded code contained the post-spike implementation used for testing purposes, rather than the pre-spike RoPE used in the paper.
> 2. RotaryEmbedding1DSpatial for RoPE (`SSARoPE` initialization): The uploaded code was configured to use `1DSpatial` RoPE instead of the final 2D RoPE method.
>
> **W1. Pre-spike vs. Post-spike**
>
> As mentioned above, the previously uploaded code was a test version using post-spike RoPE, and it has now been corrected to the pre-spike version. Applying RoPE, which involves rotating vectors by specific angles, to the Query and Key in a post-spike state destroys the inherent discrete nature of SNNs. Furthermore, applying RoPE to low-resolution binary spikes introduces severe information distortion. Indeed, our comparative experiments confirmed that the post-spike method resulted in an $R^2$ of 0.677, a significant degradation compared to the baseline performance. In contrast, our proposed pre-spike method applies rotation to the continuous membrane potentials prior to LIF binarization. This approach improves the precision of the rotation operation, and as proven in Theorem 1 of our paper, the relative phase information is preserved as a statistical expectation even after passing through the nonlinearity of LIF neurons, achieving consistent performance improvements. The experimental results are detailed in Appendix H.
> |                                 |        | Metr-la (L = 12) |      |      |      | Pems-bay (L = 12) |      |      |      | Solar (L = 168) |      |      |      | Electricity (L = 168) |      |      |      |      |
> |---------------------------------|:------:|:----------------:|:----:|:----:|:----:|:-----------------:|:----:|:----:|:----:|:---------------:|:----:|:----:|:----:|:---------------------:|:----:|:----:|:----:|:----:|
> | Models                          | Metric |         6        |  24  |  48  |  96  |         6         |  24  |  48  |  96  |        6        |  24  |  48  |  96  |           6           |  24  |  48  |  96  | Avg. |
> | Spikformer w/Conv-PE            |   R^2  |       .713       | .527 | .399 | .267 |        .773       | .697 | .686 | .667 |       .929      | .828 | .744 | .674 |          .959         | .955 | .955 | .954 | .733 |
> |                                 |   RSE  |       .565       | .725 | .818 | .903 |        .514       | .594 | .606 | .621 |       .272      | .426 | .519 | .586 |          .373         | .371 | .379 | .382 | .541 |
> | Spikformer w/SF-PE (Post-spike) |   R^2  |       .558       | .358 | .281 | .145 |        .682       | .669 | .655 | .646 |       .881      | .791 | .735 | .689 |          .941         | .939 | .932 | .929 | .677 |
> |                                 |   RSE  |       .702       | .846 | .895 | .976 |        .591       | .602 | .615 | .623 |       .355      | .470 | .529 | .573 |          .431         | .439 | .454 | .367 | .592 |
> | Spikformer w/SF-PE (Pre-spike)  |   R^2  |       .739       | .561 | .432 | .317 |        .783       | .713 | .698 | .670 |       .939      | .877 | .782 | .752 |          .981         | .975 | .972 | .965 | .760 |
> |                                 |   RSE  |       .538       | .698 | .795 | .871 |        .499       | .576 | .593 | .618 |       .251      | .362 | .479 | .511 |          .240         | .280 | .300 | .336 | .497 |
>
> **W2. Linear assumption for firing probability function**
>
> We conducted an extensive empirical analysis of the pre-spike membrane potential distributions and the corresponding approximation errors throughout the training process. Our findings, detailed in Appendix B, provide strong evidence for the validity of our assumption. As shown in Figure 3, Batch Normalization effectively stabilizes the pre-spike membrane potentials of Query and Key. The distributions are consistently zero-centered ($\mu \approx 0$) with a narrow spread ($\sigma \approx 1.1$). Crucially, over 95% of the input data falls within the range $[-2, 2]$. Figure 4 demonstrates that within this dominant input range ($[-2, 2]$), the actual surrogate gradient function (ATan) and its first-order Taylor approximation are nearly indistinguishable, showing an extremely high correlation ($>0.96$). While the approximation error increases in the saturation regions ($|u| > 2$), the probability density of the input data in these regions converges to zero (sub-Gaussian tail). Since the expected error is the integral of the pointwise error weighted by the probability density, the contribution from these high-error regions is statistically negligible. Therefore, the linear approximation holds with high precision for the vast majority of effective inputs, ensuring that the phase preservation property derived in Theorem 1 remains valid in practice.

---

> ### Author Response · Authors · 2025-11-21
>
> **W3. Extended evaluation on NLP tasks**
>
> SNN training involves uncertainty in spike generation, which can lead to performance variations under identical experimental conditions or failure to converge depending on the dataset[1]. The lack of performance improvement on the RTE task in Table 2 is attributable to this phenomenon. To demonstrate the robustness of our proposed method, we added the STS-B dataset, where similar issues have been reported. SF-PE was evaluated on all English datasets used in previous PE research (MR, Subj, SST-2, SST-5) as well as four tasks from the GLUE benchmark (SST-2, MRPC, STS-B, RTE). The results demonstrate that our method yields consistent performance improvements across all tested datasets.
> |                           | STS-B(Pearson Correlation) |
> |---------------------------|:--------------------------:|
> | Spikformer w/Conv-PE      |           0.1871           |
> | Spikformer w/CPG-PE       |           0.1871           |
> | Spikformer w/SF-PE (Ours) |           0.1924           |
>
> **W4. Extended evaluation on length extrapolation**
>
> We expanded our length extrapolation experiments to METR-LA and PEMS-BAY as suggested. SF-PE showed consistently high performance on all datasets, and the corresponding analysis has been added to Section 5.3.
> |                           |        | Metr-la (L = 12 -> 168) |      |      |      | Pems-bay (L = 12 -> 168) |      |      |      | Solar (L = 12 -> 168) |      |      |      | Electricity (L = 12 -> 168) |      |      |      |      |
> |---------------------------|:------:|:----------------------------------:|:----:|:----:|:----:|:-----------------------------------:|:----:|:----:|:----:|:--------------------------------:|:----:|:----:|:----:|:--------------------------------------:|:----:|:----:|:----:|:----:|
> | Models                    | Metric |                  6                 |  24  |  48  |  96  |                  6                  |  24  |  48  |  96  |                 6                |  24  |  48  |  96  |                    6                   |  24  |  48  |  96  | Avg. |
> | Spikformer w/CPG-PE       |   R^2  |                .551                | .339 | .307 | .149 |                 .677                | .631 | .594 | .529 |               .928               | .747 | .513 | .342 |                  .979                  | .975 | .967 | .961 | .637 |
> |                           |   RSE  |                .708                | .859 | .879 | .974 |                 .595                | .636 | .675 | .739 |               .273               | .512 | .741 | .918 |                  .266                  | .284 | .321 | .344 | .608 |
> | Spikformer w/SF-PE (Ours) |   R^2  |                .601                | .387 | .257 | .187 |                 .694                | .679 | .653 | .647 |               .936               | .764 | .528 | .371 |                  .980                  | .977 | .971 | .966 | .662 |
> |                           |   RSE  |                .667                | .827 | .910 | .952 |                 .579                | .593 | .617 | .622 |               .256               | .493 | .756 | .889 |                  .263                  | .279 | .302 | .324 | .583 |

---

### Official Review · Reviewer_nkFe · 2025-10-31

**Soundness:** 2
**Presentation:** 3
**Contribution:** 2
**Rating:** 2
**Confidence:** 2

**Summary:**

The paper presents a novel method that combines Spiking-RoPE with CPG positional encoding in spiking neural networks, resulting in a fused position encoding mechanism called Spiking Fused-PE. The approach first injects absolute positional information from CPG-PE into token embeddings via linear projection, then applies a two-dimensional Spiking-RoPE rotation to encode relative positions along both spatial (sequence) and temporal dimensions. Following this 2D rotation, the attention computation is decomposed into amplitude terms carrying absolute information and trigonometric terms encoding relative positional differences. The goal of this design is to jointly capture absolute and relative positions across space and time, yielding richer and more structured spatiotemporal representations. Experiments demonstrate the effectiveness of Spiking Fused-PE through quantitative performance gains and ablation studies.

**Strengths:**

This paper extends the RoPE mechanism to the temporal dimension of spiking neural networks and integrates it with CPG absolute positional encoding to form a new positional encoding scheme. This combination is somewhat novel. Additionally, the paper provides detailed mathematical derivations, and includes visual illustrations of the Spiking-RoPE computation process in the appendix, which together enhance clarity and facilitate understanding.

**Weaknesses:**

1) The paper’s overall innovation appears somewhat limited. It mainly extends RoPE to the time-step dimension and then merges it with CPG positional encoding, which results in a method that integrates existing ideas rather than introducing a fundamentally new mechanism. Although the proposed Spiking Fused-PE achieves good performance, it lacks deeper methodological novelty or theoretical breakthroughs.

2) The paper does not explore other potential fusion strategies between CPG and Spiking-RoPE. For instance, it remains unclear whether the CPG component must be injected only at the input embedding stage or if it could be integrated after the 2D RoPE rotation or within later layers, which might lead to different positional interaction effects.

3) The experimental section lacks sufficient discussion of computational cost. Specifically, Table 2 reports identical inference times across all compared methods, including SF-PE, which theoretically requires additional computation for the 2D RoPE operation. This inconsistency raises concerns about the rigor of the experimental setup and whether the runtime measurements were properly controlled or averaged.

**Questions:**

1) Could the authors explore whether CPG can be applied at other stages of the model. For example, after the 2D RoPE rotation or within later processing blocks rather than being limited to the input x? Would such variations affect the encoding of absolute and relative positional information?

2) In Table 2, why are the inference times completely identical for all three methods? Since SF-PE introduces additional 2D RoPE computations on top of CPG-PE, one would expect a measurable increase in inference time. How were these runtimes obtained and were they averaged over multiple runs, and was the hardware or implementation optimized in a way that masked the computational overhead?

---

> ### Author Response · Authors · 2025-11-21
>
> We appreciate your valuable reviews. We would like to address the issues found in the code uploaded in the previous submission.
>
> 1. Post-spike RoPE in SSA (`Spikformer_cpg_rope.py`, `forward` method in `SSARoPE`): The uploaded code contained the post-spike implementation used for testing purposes, rather than the pre-spike RoPE finally used in the paper, which has been corrected.
> 2. RotaryEmbedding1DSpatial for RoPE (`Spikformer_cpg_rope.py`, `SSARoPE` initialization): The uploaded code was incorrectly configured to use `1DSpatial` RoPE instead of the final 2D RoPE method, which has been corrected.
>
> We believe these discrepancies may have prevented our contributions from being fully recognized. Therefore, we have restated the paper’s contributions and novelty under these corrections:
> Our proposed method is the first to apply RoPE in SNNs by proving that the effect of RoPE is preserved in LIF activity. We explicitly model the intrinsic spatiotemporal characteristics of SNNs by proposing 2D RoPE, which independently rotates the sequence and time axes. We introduce SF-PE, a Fused-PE scheme that combines this relative PE (2D RoPE) with the SOTA absolute PE method (CPG-PE), demonstrating consistent performance improvements in both time-series forecasting and NLP tasks.
> In particular, through this rebuttal, we have strengthened our contributions further and demonstrated the effectiveness of our method more extensively by adding experiments on computational overhead, the STS-B task in NLP, extended length extrapolation, and validation of the linear approximation assumption for the spike firing function in Sec. 4.2.3.
>
> **W1. Novelty**
>
> Relative PE methods generally have sinusoidal characteristics that are incompatible with the LIF mechanism (binarization) of SNNs, limiting their direct application. RoPE, specifically, had not been applied to SNNs due to its vector rotation operations. Accordingly, we demonstrated via Sec. 4.2.3 and Appendix A that the effect of RoPE is preserved even after spike binarization, showing for the first time that RoPE can be used in SNNs. We also proposed Fused-PE, combining the independently used CPG and RoPE, proved the complementary nature of absolute-relative PE fusion in Sec. 4.3, and visually demonstrated the method's effectiveness in Fig. 3.
>
> **W2, Q1. Fusion strategy**
>
> CPG is an absolute PE method that generates positional information vectors corresponding to the input data length. Since the dimensions of Q and K are already expanded, applying CPG at this stage would require using expanded input sequences (in this case, 256x256), resulting in excessive training costs that are not practically feasible.
>
> **W3, Q2. Efficiency comparison**
>
> The "Param" column in Table 2 indicates the number of model parameters; the proposed SF-PE method is a module applied without adding any extra parameters. Separately, we have added comparisons of GFLOPs and training time in Appendix G, showing that the additional computational overhead of SF-PE is negligible.
>
> |                                                      | Params(M) | GFLOPs | Avg training time per epoch(sec) | Avg inference time per epoch(sec) |
> |------------------------------------------------------|:---------:|:------:|:--------------------------------:|:---------------------------------:|
> | Spikformer w/Conv-PE                                 |    1.67   |  72.07 |               42.56              |                5.16               |
> | Spikformer w/CPG-PE                                  |    1.7    |  72.08 |               42.67              |                5.23               |
> | Spikformer w/SF-PE (Ours) |    1.7    |  72.09 |               42.74              |                5.26               |

---

### Author Response · Authors · 2025-11-30

Dear Area Chair,

Below we summarize the reviewers’ key concerns **explicitly identifying which reviewer raised each issue** and how we addressed them.

**All fixes, experiments, and clarifications are reflected in the revised manuscript and updated code.**

## 1. Code / Implementation Mismatch

**Concern (Reviewers LdHq**, **zk6P**, and **nkFe):**

All three reviewers independently observed that the uploaded code applied **post-spike RoPE** and **1D RoPE**, which contradicted the manuscript’s claim that experiments used **pre-spike, 2D Spiking-RoPE fused at Q/K**. This raised doubts about whether the results matched the described method, and whether RoPE can even work on binary spikes.

**Our Resolution:**

- The uploaded code was an incorrect **internal test version**.
- We replaced it with the **correct pre-spike, 2D Spiking-RoPE implementation** used for all results.
- We added a **pre vs. post-spike ablation** (Appendix H):
    - **Post-spike RoPE** severely degrades performance (avg R² ≈ 0.677).
    - **Pre-spike RoPE** matches the theory and yields consistent gains.

        This resolves the implementation mismatch and validates that all reported results use the intended method.


---

## 2. Validity of the Linear Approximation Assumption

**Concern (Reviewer zk6P):**

The theoretical phase-preservation result relies on the LIF firing surrogate being approximately linear in the operating range. The reviewer asked whether this linear regime holds in practice and whether approximation errors were quantified.

**Our Resolution (Appendix B):**

- BatchNorm keeps membrane potentials **zero-centered with narrow variance**, and **>95%** of values lie in a region where ATan and its first-order approximation are nearly identical (correlation > 0.96).
- Saturation regions have negligible density.

    Thus, the model predominantly operates in the linear regime assumed by our theory, making the approximation practically valid.


---

## 3. Novelty & Fusion Strategy

**Concern (Reviewer nkFe):**

The reviewer felt the method might be “incremental”—extending RoPE temporally and fusing it with CPG-PE—and asked whether other fusion placements were considered.

**Our Resolution:**

- Prior work had **not** shown that RoPE is compatible with SNNs; we are the **first to prove and demonstrate** that RoPE remains valid under LIF *when applied pre-spike*.
- We introduced **2D Spiking-RoPE**, which separately rotates sequence/time axes and explicitly models spatiotemporal structure.
- We showed that SF-PE produces complementary **absolute (row/column)** vs **relative (diagonal)** attention structures, expanding expressiveness.
- Alternative fusion placements (e.g., after 2D RoPE or deeper in the stack) would require operating on **expanded 2D sequences**, which is computationally prohibitive in SNNs; we identify these as future directions.

---

## 4. Runtime, Task-Specific Behavior, and Length Extrapolation

### (a) Runtime measurement

Concern (**Reviewer nkFe)**: Table 2 showed identical inference times despite additional 2D RoPE computation, raising doubts about runtime methodology.

**Response:** We added **GFLOPs, training time, and inference time** per epoch (Appendix G). SF-PE adds only **negligible** overhead (differences in the third decimal place).

### (b) NLP task behavior (RTE)

Concern (**Reviewer zk6P)**: CPG-PE and SF-PE did not improve RTE, and no explanation was provided.

**Response:** We clarified that spike-based models show **dataset-dependent instability**. To verify robustness, we added **STS-B** and evaluated across **MR, Subj, SST-2/5, MRPC, STS-B, RTE**. SF-PE shows **consistent improvements** across all tasks, including the newly added STS-B.

### (c) Length extrapolation presentation

Concern (**Reviewer zk6P)**: The extrapolation claims were mostly relegated to the appendix.

**Response:** We expanded length extrapolation to **METR-LA** and **PEMS-BAY**, and moved the core results into **Section 5.3**. SF-PE consistently outperforms CPG-PE when inference lengths exceed training lengths.

---

### Closing

The reviewers’ concerns centered on:

1. **Implementation mismatch** (LdHq, zk6P, nkFe)
2. **Theoretical assumption validity** (zk6P)
3. **Novelty and fusion design** (nkFe)
4. **Runtime, NLP behavior, and extrapolation** (nkFe, zk6P)

We have addressed each point with corrected code, new ablations, strengthened theoretical support, and expanded experiments.

We respectfully request that these clarifications be considered in the final assessment.

---

### Meta-Review · Area_Chair_McS5 · 2026-01-05

**Summary:**

Reviewers expressed concern that, while the paper addresses an important problem of positional encoding distortion in spiking Transformers and explores an interesting direction by adapting RoPE to spiking dynamics, the overall conceptual novelty is limited. Several reviewers viewed the proposed method as primarily an integration and adaptation of existing positional encoding techniques (RoPE and CPG-based absolute PE) to the spiking setting, rather than the introduction of a fundamentally new positional encoding or attention mechanism.

Reviewers also expressed significant concern regarding the correctness and reproducibility of the implementation. Multiple reviewers initially identified discrepancies between the method described in the paper and the released code (e.g., pre-spike vs. post-spike RoPE and 2D vs. 1D implementations). Although the authors stated in their rebuttal that these issues were fixed and that all experiments used pre-spike 2D RoPE,  inspected the updated code myself, and I still believe there are implementation issues that may prevent the intended RoPE computation from executing as described, potentially causing silent fallback behavior.

Reviewers further expressed concern about the strength and generality of the theoretical assumptions, particularly the reliance on approximate linearity of the LIF firing function.

Finally, reviewers noted that the empirical improvements are modest and task-dependent, with limited evidence of scalability or clear advantage over existing positional encoding approaches. Taken together, these concerns led reviewers to conclude that the paper does not yet meet the required bar in novelty, methodological clarity, and experimental reliability.

**Reviewer Concerns:**

**Concerns partially addressed by the rebuttal**: Reviewers’ questions regarding the validity of the theoretical assumptions were partially addressed by additional empirical analysis showing that the LIF firing function operates largely in a near-linear regime during training. The rebuttal also added runtime and computational cost analyses, clarifying that the overhead of the proposed method is small. In addition, concerns about task coverage and length extrapolation were partially mitigated through added NLP tasks and expanded extrapolation experiments.

**Concerns that remain outstanding**: Reviewers’ concerns about limited conceptual novelty remain largely unresolved, as the rebuttal does not substantially change the perception that the method primarily integrates existing positional encoding techniques. More critically, concerns regarding implementation correctness and reproducibility remain outstanding: despite claims that the code was fixed, my own inspection of the updated code reveals issues that may prevent the intended pre-spike 2D RoPE computation from executing as described, continuing to undermine confidence in the experimental results. Finally, the modest and task-dependent empirical gains are unlikely to convince reviewers that the proposed method offers a clear advantage over existing approaches.

**Reviewer Scores:**

- **Reviewer LdHq (Score: 2 – Reject)**: This reviewer’s main concerns centered on discrepancies between the paper and the released code, particularly regarding pre-spike vs. post-spike RoPE and 2D vs. 1D implementations. Although the rebuttal claimed these issues were fixed, subsequent inspection suggests implementation problems remain. As a result, it is unlikely this reviewer would have increased their score; their assessment would most likely remain unchanged.

- **Reviewer nkFe (Score: 2 – Reject)**: This reviewer expressed concerns about limited novelty, unclear fusion strategy exploration, and questionable runtime reporting. While the rebuttal addressed runtime measurement and added overhead analysis, it did not materially change the novelty concerns. At best, the reviewer might have increased the score to 4.

- **Reviewer zk6P (Score: 4 – Marginally below threshold)**: This reviewer was the most borderline positive and raised concerns about theoretical assumptions, task-specific behavior, and presentation of length extrapolation. The rebuttal provided additional empirical validation and expanded experiments, which may have alleviated some concerns. However, given the remaining issues regarding implementation reliability and modest gains, this reviewer would likely maintain their original score rather than move to acceptance.

---

### Decision · Program_Chairs · 2026-01-26

Reject